# Recently activated CD4 T cells in tuberculosis express OX40 as a target for host-directed immunotherapy

Abigail R. Gress[1,2], Christine E. Ronayne[1,2], Joshua M. Thiede ®[1,2], David K. Meyerholz ®[3], Samuel Okurut ®[4], Julia Stumpf[1], Tailor V. Mathes ®[1,2], Kenneth Ssebambulidde ®[4], David B. Meya[4], Fiona V. Cresswell ®[4,5,6], David R. Boulware ®[1] & Tyler D. Bold ®[1,2] ✉

After Mycobacterium tuberculosis (Mtb) infection, many effector T cells traffic to the lungs, but few become activated. Here we use an antigen receptor reporter mouse (Nur77-GFP) to identify recently activated CD4 T cells in the lungs. These Nur77-GFP[HI] cells contain expanded TCR clonotypes, have elevated expression of co-stimulatory genes such as *Tnfrsf4*/OX40, and are functionally more protective than Nur77-GFP[LO] cells. By contrast, Nur77-GFP[LO] cells express markers of terminal exhaustion and cytotoxicity, and the trafficking receptor *S1pr5*, associated with vascular localization. A short course of immunotherapy targeting OX40[+] cells transiently expands CD4 T cell numbers and shifts their phenotype towards parenchymal protective cells. Moreover, OX40 agonist immunotherapy decreases the lung bacterial burden and extends host survival, offering an additive benefit to antibiotics. CD4 T cells from the cerebrospinal fluid of humans with HIV-associated tuberculous meningitis commonly express surface OX40 protein, while CD8 T cells do not. Our data thus propose OX40 as a marker of recently activated CD4 T cells at the infection site and a potential target for immunotherapy in tuberculosis.

Tuberculosis (TB) is a leading infectious cause of death worldwide, and an estimated one-quarter of the global population is infected[1]. Despite the widespread prevalence and health burden of TB, we still lack efficacious vaccines and efficient drug treatment regimens. The current antibiotic regimen for TB disease is months-long, contributing to multiple challenges of successful TB treatment[2]. One opportunity for improved TB drug development is to harness the potential of the adaptive immune response, specifically targeting CD4 T cells, which play an essential protective role in both mouse models[3,4] and humans with TB[5].

During the adaptive immune response against *Mycobacterium tuberculosis* (*Mtb*), millions of CD4 T cells traffic to the site of infection, but relatively few find antigen and become activated to carry out effector functions[6,7]. Prior studies have identified various phenotypes of CD4 T cells recruited to the lungs of *Mtb*-infected mice, based on lung compartmental localization, expression of transcription factors and surface markers, and secretion of effector cytokines[8–11]. CD4 T cells specific for a highly abundant *Mtb* antigen, ESAT-6, frequently express the surface markers CX3CR1 and KLRG1 and secrete IFN-γ, but primarily localize to the lung vasculature[8,9]. These terminally

[1]Department of Medicine, University of Minnesota, 420 Delaware Street, SE MMC 250, Minneapolis, MN 55455, USA. [2]Center for Immunology, 2101 6th St SE, WMBB 2-118, University of Minnesota, Minneapolis, MN 55455, USA. [3]Department of Pathology, Roy J. and Lucille A. Carver College of Medicine, 1165 Medical Laboratories (ML), 51 Newton Rd, University of Iowa, Iowa City, IA 52242, USA. [4]Infectious Diseases Institute, P.O. Box 22418, Makerere University, Kampala, Uganda. [5]MRC/UVRI and London School of Hygiene and Tropical Medicine Uganda Research Unit, PO Box 49, Plot 51-59, Nakiwogo Road Entebbe, Entebbe, Uganda. [6]Department of Global Health and Infection, Brighton and Sussex Medical School, Brighton, East Sussex BN1 9PX, UK. ✉e-mail: tbold@umn.edu

differentiated, vascular, Th1 effector cells are less protective and phenotypically distinct from CD4 T cells expressing CXCR3 that localize to the lung parenchyma[8,9]. However, the diverse expression profiles of these subsets have not been broadly defined in relation to their activation status in vivo, due to reliance on targeted flow cytometry panels and intracellular cytokine staining, a technique that requires fixation preventing downstream transcriptional profiling and live cell assays.

In this study, we used the Nur77-GFP mouse, which provides temporally-restricted, antigen-dependent green fluorescent protein (GFP) expression after T cell receptor (TCR) stimulation[12], to discriminate recently activated *Mtb*-specific CD4 T cells (Nur77-GFP[HI]) and not recently activated cells (Nur77-GFP[LO]). We hypothesize that activated CD4 T cells coordinate diverse effector mechanisms during TB that could be harnessed for development of host-directed immunotherapies. Using traditional targeted assays and high-resolution bulk and single-cell transcriptomics, we find that activated CD4 T cells at the site of *Mtb* infection encompass multiple phenotypic types that express an array of costimulatory markers, including OX40 and are functionally superior to non-activated cells. By studying cells recruited to the site of human TB meningitis infection, we observe that OX40 expression is conserved across species among a sub-population of effector CD4 T cells bearing expression hallmarks of antigen receptor activation. Targeting OX40+ cells with an agonist monoclonal antibody during early stages of mouse *Mtb* infection results in a lasting immunotherapeutic benefit, in contrast to the adverse impacts of PD-1 deficiency or blockade[13–16]. Our study adds an additional approach to others demonstrating the success of T cell-targeted immunotherapies on improving TB outcomes in mice[6,17–19].

## Results

### CD4 T cells express Nur77-GFP with antigen recognition in *Mtb* infection

We first characterized Nur77-GFP reporter expression by T cells in the lungs after aerosol infection with Mtb. Among CD44+ effector CD4+ T cells in the lungs, we observed a continuum of Nur77-GFP expression ranging between 10-25% Nur77-GFP[HI] between individuals and experiments (Fig. S1a and b). This suggested a range of TCR signal strength among effector T cells and asynchronous antigen recognition with transient expression of the reporter, which becomes downregulated within 48 hours after TCR stimulus[12]. We found that Nur77-GFP expression was more common among CD44+ effector CD4 T cells when compared to CD8 T cells, but that the activated cell population peaked at Day 28 postinfection for both cell types (Fig. S1c and d). For subsequent analyses of the Nur77-GFP mouse we focused on Day 28, a time point when robust T cell responses contribute to curtailing *Mtb* growth in vivo, similar to what has been observed in wild type C57BL/6 mice (Fig. S1c)[4]. At this time point, infected Nur77-GFP mice had a higher percentage of effector CD4 T cells expressing Nur77-GFP than uninfected Nur77-GFP, uninfected wildtype, and infected wildtype mice (Fig. 1a), consistent with T cell activation in response to *Mtb*. To confirm that Nur77-GFP expression is due to direct, recent *Mtb* antigen recognition instead of antigen-independent activation, we included injection of the *Mtb* peptide Ag85B$_{240-254}$ six hours prior to harvest. Ag85B$_{240-254}$ peptide injection resulted in a clear increase in Nur77-GFP expression in only the Ag85B-specific CD4 T cells (Fig. 1b). These results suggest that Nur77-GFP mice can discriminate *Mtb*-specific T cells recently activated by antigen receptor stimulation (Nur77-GFP[HI]) from other (Nur77-GFP[LO]) CD4 T cells not recently activated and determine what distinguishes these populations.

### Nur77-GFP[LO] and Nur77-GFP[HI] CD4 T cells are phenotypically distinct but overlapping

To compare these populations, we limited all our subsequent analysis to CD44+ CD4 T cells, because CD44 expression implies that cells previously received antigen receptor stimulation leading to clonal expansion and differentiation into effector cells. By focusing on CD44+ cells, it is therefore less likely that Nur77-GFP[LO] population represents cells that are not *Mtb*-specific. Instead, we hypothesized that most Nur77-GFP[LO] CD44+ CD4 T cell population are *Mtb*-specific effector cells expanded upon infection, but that have not recently been activated at the time of the harvest. These cells are likely a diverse population, including those that are either: spatially isolated from their cognate antigen, specific for an antigen that is not readily available at the time of harvest, or which express T cell receptors (TCR) with relatively low affinity for an Mtb peptide.

In contrast, we hypothesized that the Nur77-GFP[HI] population is enriched for cells with particular, important phenotypic characteristics, spatial localization, and antigen specificity or affinity that increase the likelihood of antigen-specific CD4 effector T cell activation. To test this, we used flow cytometry to compare the two populations from the same mouse, comprising the highest and lowest ~1/3 of GFP expression. We chose this GFP gating strategy as an unbiased method due to the variability of GFP expression across both mice and experiments (Fig. S1a and b). Mice received an intravenous injection of fluorescent CD45 antibody immediately before harvest to distinguish cells localizing to the lung vasculature or parenchyma. Nur77-GFP[LO] CD4 T cells included a larger population of vascular cells and cells expressing CX3CR1 and KLRG1 (Fig. 1c), markers of terminally differentiated CD4 T cells in TB and other chronic bacterial infections[8,9,20]. Nur77-GFP[HI] CD4 T cells rarely localized to the vasculature, were more often CX3CR1-, KLRG1- and contained more cells recognizing the *Mtb* antigens Ag85B and ESAT-6 (Fig. 1c). These differing CX3CR1, KLRG1, and tetramer specificity phenotypes were observed not only for bulk effector CD4 T cells in the lungs, but also among the subset of lung parenchymal (IV-) cells (Fig. S1e). These data indicate that activated CD4 T cells are phenotypically distinct from not recently activated cells and identify lung compartmental localization as a key determinant of CD4 T cell activation in TB.

### Nur77-GFP[HI] cells are more functionally protective against *Mtb* infection

To compare the functional protective capacity of recently activated vs other CD4 T cells, we sorted the two populations from the lungs of infected Nur77-GFP mice and performed adoptive transfer into α/β T cell-deficient (TCRα-/-) recipients at one-week postinfection, an approach used previously to compare the relative protective capacity of lung CD4 T cell subsets in TB[13,21,22]. (Figs. 1d, S2). Both Nur77-GFP[HI] and Nur77-GFP[LO] CD4 T cell transfers extended survival compared to TCRα-/- mice receiving no adoptive transfer (median of 130 vs 70 vs 50 days, respectively Fig. 1e), but Nur77-GFP[HI] CD4 T cell transfer conferred a longer median survival time than Nur77-GFP[LO] (Fig. 1e). Both donor populations were inferior to the maximal observed potential of total bulk effector CD4 T cell population transfer ($1 \times 10^7$ cells per mouse, 236 days) (Fig. 1e). These data suggest that the population of activated Nur77-GFP[HI] CD4 T cells have superior protective capacity when compared to non-activated cells, but that each population may partially contribute to control of the infection.

We also observed differential protection by these two donor populations when endogenous α/β T cell responses are present in the recipient. We transferred either Nur77-GFP[LO] or Nur77-GFP[HI] CD4 T cell populations into infection time point matched, congenically distinct wildtype mice (CD45.1) at four weeks post-infection (Fig. 1f). Compared to mice receiving Nur77-GFP[LO] cells, Nur77-GFP[HI] recipient mice had a significantly reduced lung bacterial burden at one week after transfer (Fig. 1h). However, Nur77-GFP[LO] cells also significantly reduced bacterial load when compared to mice that did not receive a cell transfer, again suggesting that both populations contribute to infection control. Flow cytometry revealed no difference in the number of donor Nur77-GFP[LO] and Nur77-GFP[HI] CD4 T cells infiltrating the lung

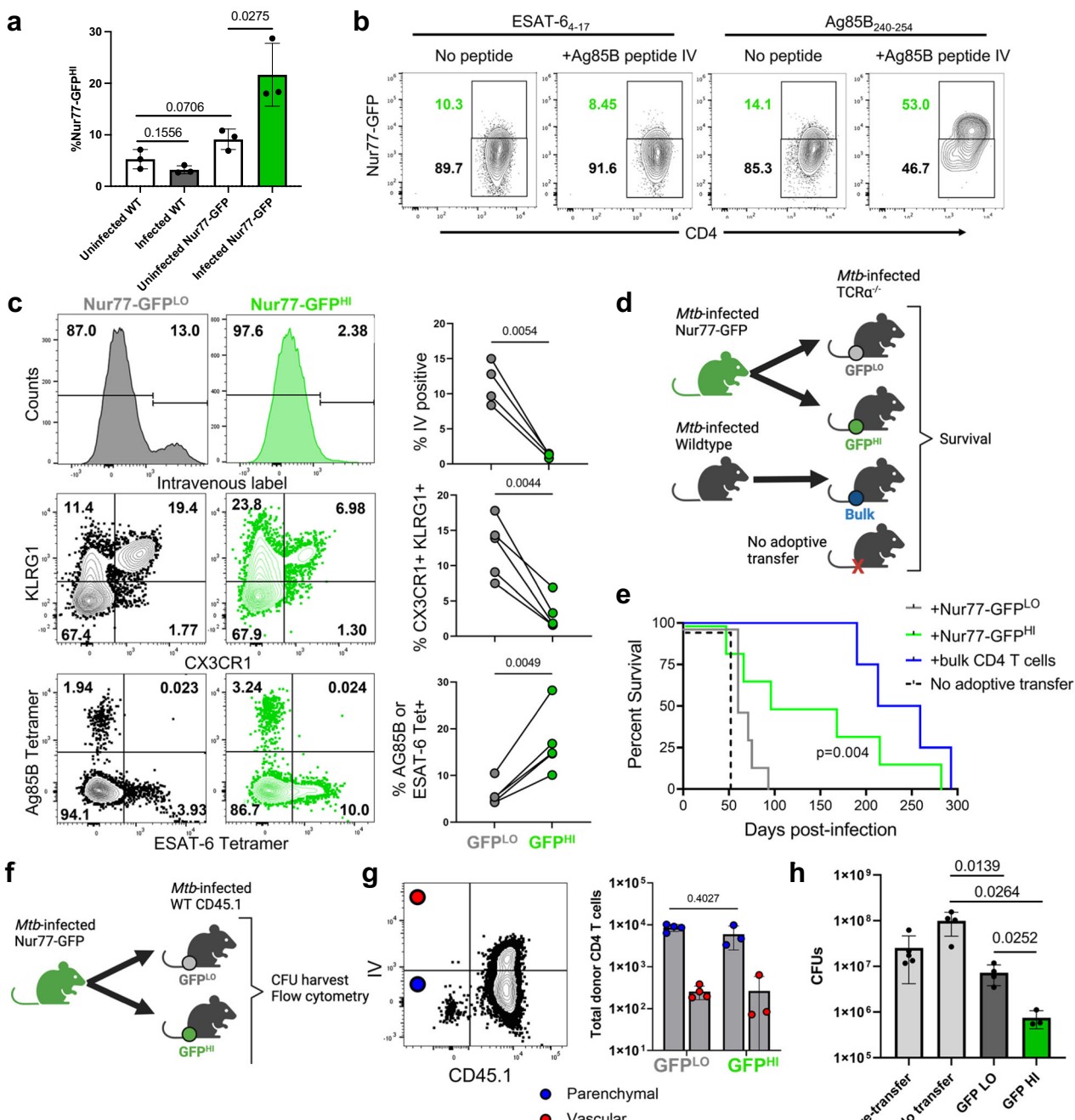

**Fig. 1 | Nur77-GFP[HI] and Nur77-GFP[LO] CD4 T cells are phenotypically and functionally distinct. a** Frequency of Nur77-GFP[HI] CD4 T cells in lungs of Nur77-GFP or wildtype mice at four weeks post-infection or uninfected. Gated on live CD4+ CD44+ Nur77-GFP[HI] cells. *P* values calculated with an unpaired two-tailed *t*-test, 3 mice per group. **b** Flow cytometry of Ag85B- or ESAT-6-specific CD4 T cells from Nur77-GFP lungs harvested at four weeks post-infection. Mice indicated received Ag85B peptide injected six hours before harvest. Gated on live CD3+ CD4+ CD44+ Tetramer+ cells. Tetramer+ cells represent either Ag85B or ESAT-6 specificity. Representative plots from experiment with 4 mice per group. **c** Flow cytometry of Nur77-GFP[LO] and Nur77-GFP[HI] CD4 T cells from Nur77-GFP lungs harvested at four weeks post-infection. CD45.2 fluorescent antibody injected intravenously immediately before harvest identifies vascular cells. Gated on live CD3+ CD4+ CD44+ and either lowest or highest 1/3 of Nur77-GFP expression. Further gated on IV-, CX3CR1+ KLRG1+, or Tetramer+ cells. Tetramer+ cells represent either Ag85B or ESAT-6 specificity. *P* values calculated with a paired two tailed *t*-test, 4 or 5 mice per group. **d** Nur77-GFP[LO] and Nur77-GFP[HI] CD4 T cells sorted from Nur77-GFP lungs at four weeks post-infection and injected into one week post-infection TCRα-/- mice. Donor cells sorted

for live CD3+ CD4+ CD44+ Nur77-GFP[LO], Nur77-GFP[HI], or total bulk CD4 T cells. **e** Curves show survival of recipient TCRα-/- mice. *P* value calculated with a two-sided Mantel-Cox test, 4-6 mice per group. **f–h** Nur77-GFP[LO] and Nur77-GFP[HI] CD4 T cells sorted from Nur77-GFP lungs at four weeks post-infection and injected into infection-matched wildtype CD45.1 mice. Donor cells sorted on live CD4+ CD44+ Nur77-GFP[LO] or Nur77-GFP[HI] cells. Recipient and no transfer wildtype CD45.1 lungs harvested at one week post-transfer for flow cytometry and lung bacterial load. **g** Number and lung localization of donor and recipient cells at one-week post-transfer. CD45.2 fluorescent antibody injected immediately before harvest identifies vascular cells. Gated on live CD4+ CD44+ cells. Cell numbers assessed using counting beads. *P* value calculated with an unpaired two tailed *t*-test. Mice per group: 4 for Nur77-GFP[LO] recipients or 3 for Nur77-GFP[HI]. **h** Lung bacterial load at one-week post-transfer or without transfer. Pre-transfer wildtype CD45.1 lungs harvested at four weeks post-infection. *P* values calculated with an unpaired two-tailed *t*-test. Mice per group: 5 for pre-transfer, 4 for no transfer, 4 for Nur77-GFP[LO] recipients or 3 for Nur77-GFP[HI]. Error bars indicate standard deviation.

parenchyma of the recipients (Fig. 1g). This suggests that the greater protection of Nur77-GFP[HI] CD4 T cells is not due to a difference in surviving cell numbers accessing the infection site, but instead to functional characteristics enriched in the activated Nur77-GFP[HI] CD4 T cell population.

## Nur77-GFP[LO] and Nur77-GFP[HI] CD4 T cells have distinct gene expression profiles

To investigate which characteristics of recently activated CD4 T cells in TB might underlie this differential protection, we performed bulk RNA sequencing (RNA-seq) on Nur77-GFP[HI] and Nur77-GFP[LO] effector CD4 T cells sorted from the lungs of *Mtb* infected mice (Supplementary Data S1). Among the differentially expressed genes that were most significantly upregulated in activated Nur77-GFP[HI] cells were costimulatory (*Tnfrsf4, Tnfrsf9*) and coinhibitory receptors (*Lag3*). Interestingly neither *Ifng* nor *Tnf* were differentially expressed at the transcriptional level, but *Il21*, a cytokine required for optimal protection against TB[23], was upregulated in recently activated cells. Despite excellent purity of Nur77-GFP[HI] and [LO] cells postsorting (Fig. S2), *Nr4a1* was not significantly differentially expressed between the two populations, likely reflecting a difference in the temporal expression pattern of Nr4a1 transcript and

Nur77GFP-reporter protein, with an expectation for shorter duration of expression of the transcript and a somewhat longer duration of the reporter protein. Consistent with our flow cytometry data, Nur77-GFP[LO] CD4 T cells differentially expressed markers of terminally differentiated cells (*Cx3cr1*), with a cytotoxic characteristic (*Gzma*), notable for expression of the transcription factors *Zeb2* and *Klf3*. (Fig. 2a and b). Nur77-GFP[LO] cells also expressed markers of signaling (*Ptpn4, Rap1b*), migration (*Ccr2, Itga4, S1pr5*), and survival (*Il7r*) (Fig. 2b). Other studies have identified *Zeb2*-mediated *S1pr5* expression as a mechanism of vascular migration among tissue-resident T cells[24], suggesting that some Nur77-GFP[LO] cells may have been previously activated in the lung parenchyma and subsequently migrated into the vasculature.

## Single-cell RNA-Sequencing reveals distinct populations of *Mtb*-specific CD4 T cells

To further interrogate the diverse expression profiles and phenotypes present in the *Mtb*-specific CD4 T cell population, we performed single-cell RNA-Sequencing (scRNA-Seq) on Ag85B- and ESAT-6-tetramer enriched[25] lung CD4 T cells from infected Nur77-GFP mice (Supplementary Data S2). The enriched *Mtb*-specific CD4 T cell population included naïve (*Ccr7*[+]), proliferating (*Mki67*[+]), and two effector (*Nkg7*[+])

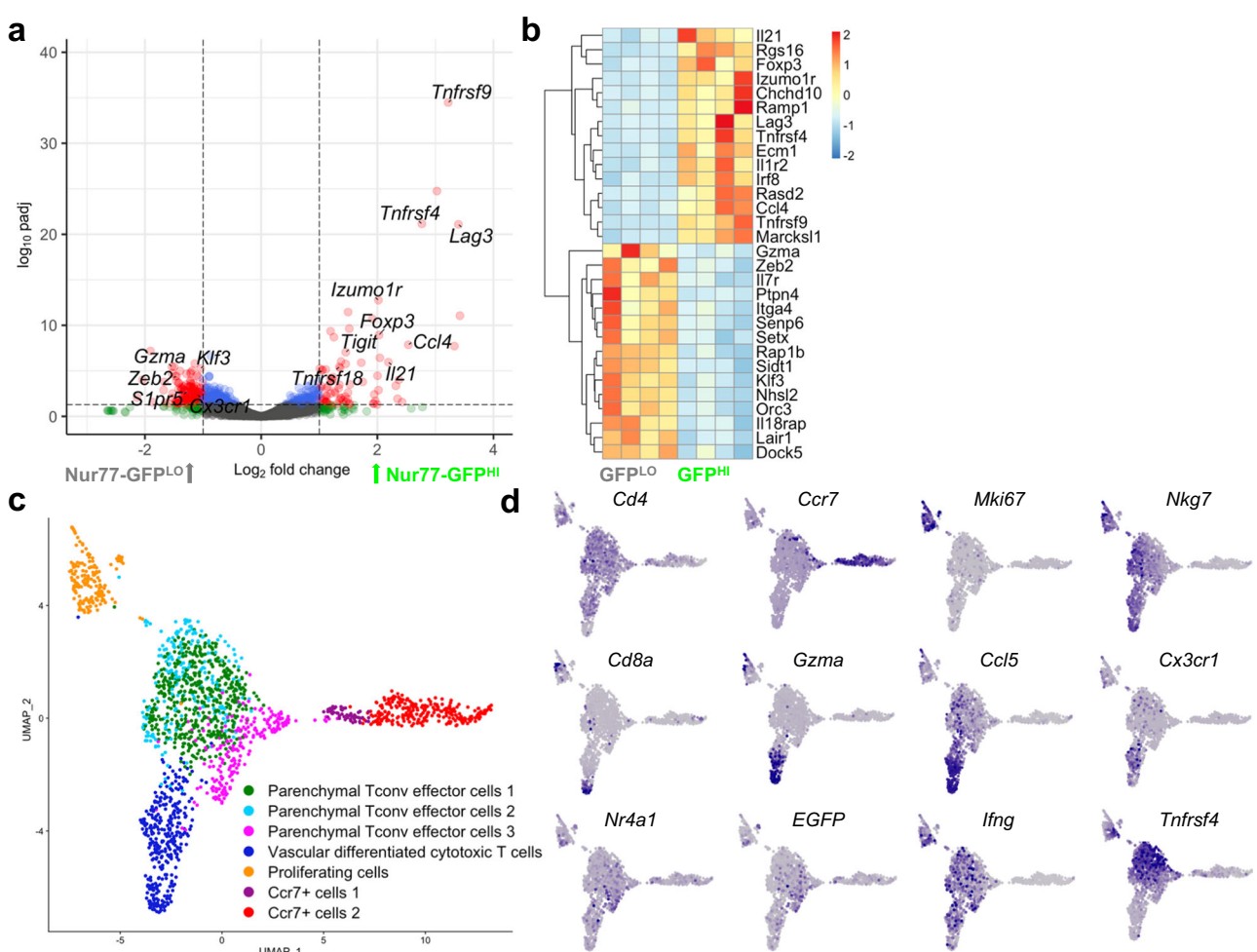

**Fig. 2 | *Mtb*-specific CD4 T cells include diverse phenotypes. a, b** Differential gene expression analysis of Nur77-GFP[LO] and Nur77-GFP[HI] CD4 T cells sorted from lungs at four weeks post-infection. Data representative of 4 biological replicates. Volcano plot highlights genes in red with log2 fold change > 1 and *p* value based on the two-tailed Wald test adjusted for multiple hypothesis testing using Benjamini-Hochberg correction with cutoff of -log padj > 1. Heatmap shows scaled differential expression between Nur77-GFP[LO] and Nur77-GFP[HI] cells of a selection of genes with -log padj > 1, with same statistical approach as above. Source data for **a,b** are located in

Supplementary Data S1. **c** UMAP projection of single-cell transcriptomes of 1,795 cells from the lungs of Nur77-GFP mice at four weeks post-infection, that underwent magnetic CD4 isolation and tetramer enrichment for Ag85B- and ESAT-6-specificity. Data represents cells pooled from 3 individual mice. Colors indicate transcriptomic clusters with manual annotation based on differential expression of marker genes. **d** Feature plots show regions of elevated expression of indicated genes. Source data for **c, d** located in Supplementary Data S2.

clusters (Fig. 2c and d). The effector clusters were split into conventional (Tconv) effector and differentiated cytotoxic (*Gzma*⁺, *Cx3cr1*⁺) clusters (Fig. 2c and d). The cytotoxic CD4 T cell cluster co-clustered with a small population of residual CD8 T cells that remained after tetramer enrichment, supporting the idea that these CD4 T cells share transcriptional programs with CD8 T cells (Fig. 2d).

*Tnfrsf4*, encoding the costimulatory receptor OX40, was highly and specifically expressed on Tconv effectors, unlike the effector cytokine *Ifng* which was broadly expressed throughout effector clusters (Fig. 2d). As expected, *Nr4a1* (encoding Nur77) and the *EGFP* reporter gene were co-expressed in cells across effector cell populations, including a subpopulation of population of cells where *Tnfrsf4* was most abundant (Fig. 2d). These data illustrate the phenotypic heterogeneity of *Mtb*-specific CD4 T cells in the lungs, potentially representing successive states of differentiation.

## Nur77-GFP^LO and Nur77-GFP^HI CD4 T cells encompass diverse populations

Among the subtypes we observed in *Mtb*-specific lung CD4 T cells, we hypothesized that differential abundance of certain phenotypes contributes to the different protective capacities of recently activated and other CD4 T cells. We therefore performed scRNA-Seq on populations of CD44⁺ effector Nur77-GFP^LO or Nur77-GFP^HI effector CD4 T cells sorted from the lungs of the same infected mouse and compared the phenotypic composition of each group (Supplementary Data S3 and S4). To identify vascular cells, we used an oligonucleotide labeled CD45 antibody injected intravenously prior to harvest. Nur77-GFP^LO and Nur77-GFP^HI samples included similar frequencies of proliferating cells (*Mki67*⁺; 3.0% vs 2.2%), but vascular differentiated cytotoxic T cells (*Gzma*⁺, *Cx3cr1*⁺, *S1pr5*+, IV⁺) were more frequent in the Nur77-GFP^LO sample (11.7% vs 5.8%). Parenchymal T cells (*Cx3cr1*⁻, IV⁻) were slightly less prevalent in the Nur77-GFP^LO sample compared to Nur77-GFP^HI (85.3% vs 92.0%) (Fig. 3a and b). These data suggest that all effector cell phenotypes observed are capable of being activated during infection. Interestingly, the Nur77-GFP^HI population also includes a cluster of cells (4.7%) marked by the expression of *Foxp3*, consistent with regulatory T cells (Tregs) (Fig. 3a and b), suggesting that this phenotype commonly expresses Nur77-GFP in the periphery and is activated by antigen receptor stimulation in the lungs.

## Nur77-GFP^HI CD4 T cells have increased expression of costimulatory molecules

Although we observed overlap between the composition of the Nur77-GFP^LO and Nur77-GFP^HI CD4 T cell populations, there were clear distinctions between the expression profiles of their effector cells. As seen in our bulk RNA-seq and tetramer-enriched scRNA-Seq experiments (Fig. 2a and d), *Tnfrsf4* was more commonly expressed by activated effector cells (Fig. 3b, Supplementary Data S4) (22% GFP^LO vs 59% GFP^HI), along with other costimulatory and coinhibitory marker genes (e.g. *Tnfrsf18*, *Tnfrsf9*, *Ctla4*) (Fig. 3c and d, Supplementary Data S4). We used flow cytometry to confirm that OX40 surface protein expression is significantly enriched in Nur77-GFP^HI CD4 T cells, in both the total effector CD4 population, as well as the subset of parenchymal (IV⁻) *Mtb*-tetramer staining population (Fig. 3e). The exception to this finding is Nur77-GFP^HI vascular effector T cells, which express relatively low levels of costimulatory and inhibitory receptors (Fig. 3c). The increased expression of costimulatory and inhibitory receptors in Nur77-GFP^HI CD4 T cells further suggests that these receptors may contribute to protection against TB.

## Single-cell TCR sequencing reveals preferential activation of expanded clonotypes

Using single-cell TCR sequencing, we tested the hypothesis that Nur77-GFP^HI CD4 T cells are more protective than Nur77-GFP^LO cells due to

differential *Mtb* antigen specificity of the two populations. The proportion of the most highly expanded TCR clonotypes was similar between Nur77-GFP^LO and Nur77-GFP^HI cells (Fig. 4a), likely reflecting the transient nature of Nur77-GFP expression- within 6-48 hours of antigen receptor stimulation and suggesting that TCR specificity is not the sole explanation for better protective capacity of Nur77-GFP^HI CD4 T cells. However, we found that the top three most highly expressed clonotypes in both populations (1, 2, and 3) were over-represented in the Nur77-GFP^HI population (Fig. 4b), suggesting that T cells with these specificities are more likely to be activated. By overlaying the distribution of each expanded clonotype on gene expression clusters for Nur77-GFP^LO cells, we found that all three clonotypes were present in both vascular and parenchymal clusters. However, among activated Nur77-GFP^HI cells, clonotypes 2 and 3 were only present in the parenchymal Nur77-GFP^HI effector cell clusters (Fig. 4c), suggesting that T cells of these specificities are more likely to bear a parenchymal Tconv cell phenotype.

## OX40 agonism of CD4 T cells in *Mtb*-infected mice results in improved outcomes

*Tnfrsf4*/OX40 was among the most highly expressed markers in bulk-activated Nur77-GFP^HI CD4 T cells and *Mtb*-specific CD4 T cells (Figs. 2a, b and 3c, d). Although OX40 contributes to CD8 T cell protection in mouse infection models with some viruses and Listeria[26–30], we found that it is predominantly expressed by CD4 T cells and rarely by CD8 T cells in the lungs after *Mtb* infection (Fig. 5a). Because it is highly expressed by activated CD4 T cells, we hypothesized that OX40 stimulation may contribute to their enhanced protective capacity. To test this hypothesis, we examined the impact of enhanced OX40 stimulation on *Mtb* infection outcomes, using an agonist monoclonal antibody.

We found that a two-week course of OX40 agonist antibody treatment at the first month of infection extended long-term survival of wild type mice (Figs. 5b and S3a) and had no apparent detrimental impact on host health (Fig. S3b). This was in stark contrast to treatment with a PD-1 blocking antibody, which resulted in several cases of early mortality after treatment and no extension of survival compared to isotype control (Fig. S3c). Wild-type mice that received OX40 agonist treatment for two weeks also had a ~50% reduction in lung bacterial load compared to isotype control and PD-1 blocking antibody (Fig. 5c). These data are consistent with existing evidence regarding the detrimental effect of PD-1 deficiency or blockade in tuberculosis[13,14] and suggest that different approaches to T cell targeted immunotherapy can have divergent effects on *Mtb* infection outcomes.

Notably, OX40 agonist treatment did not affect survival of TCRα-/- mice (Fig. S3a), suggesting that this beneficial effect is mediated by α/β T cells. OX40 agonism survival extension after only two weeks of treatment during early infection suggests that this treatment causes long-term changes in CD4 T cells that can continue to improve infection outcomes months after treatment has ended.

## OX40 agonism alters lung inflammation

To determine whether the change in survival was accompanied by changes in structural disease, we analyzed histopathology in OX40 agonist, PD-1 blockade, and isotype control-treated lungs. We found in all treatment groups that regions of inflammation were patchy to coalescing (Fig. S4a). However, OX40 agonism-treated lungs had more extensive inflammation characterized primarily by multifocal to coalescing cellular aggregates that were on average larger in size than those from mice receiving isotype control or PD-1 blockade (Fig. S4a–c). Although cellular inflammation in all samples was principally composed of either macrophages or lymphoid cells, due to the coalescing nature of cellular inflammation in OX40 agonist-treated mice, these lungs had less discrete macrophage and lymphoid aggregates (Fig. S4c). In addition OX40 agonist-treated lungs were further

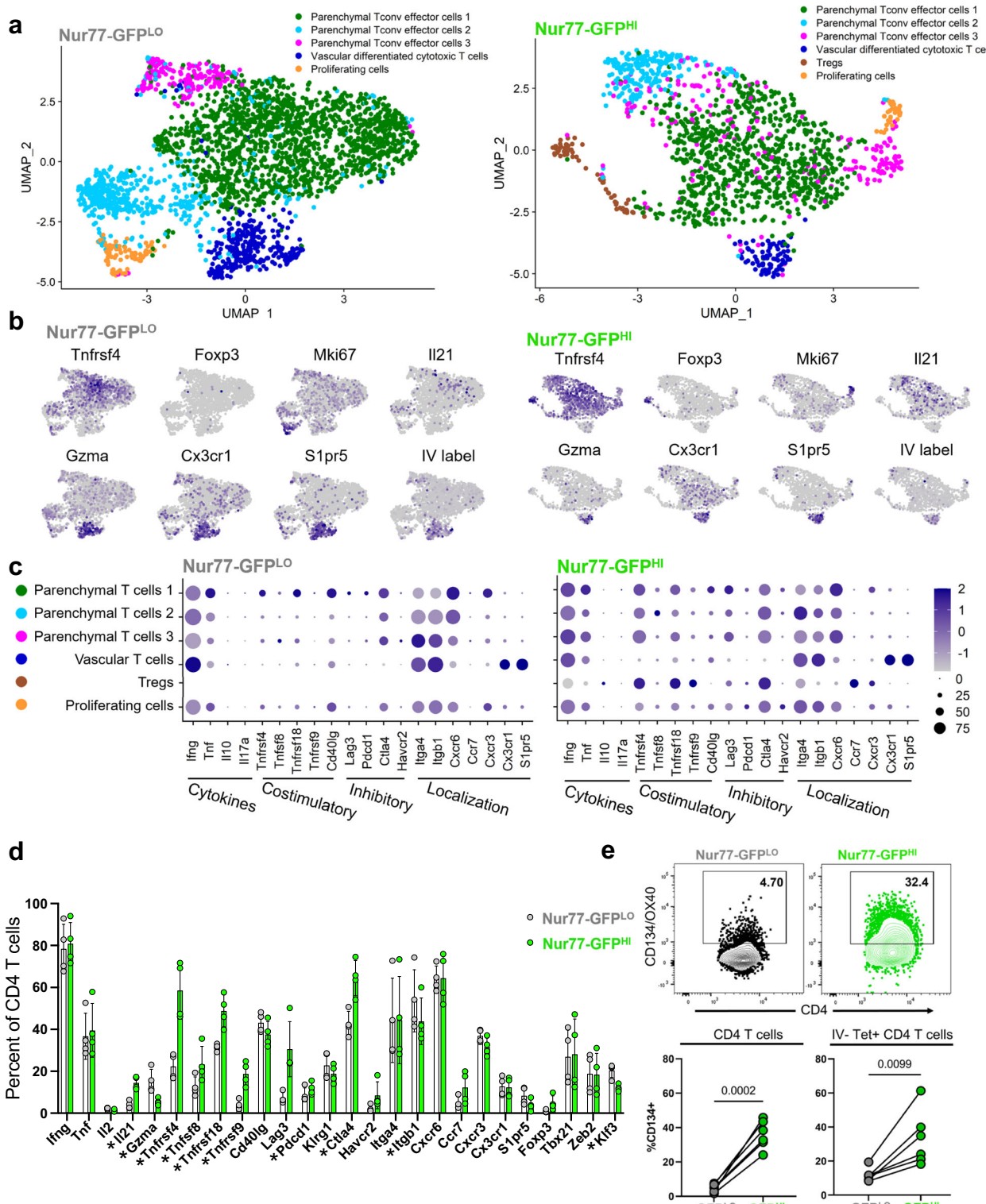

**Fig. 3 | Nur77-GFP^HI and Nur77-GFP^LO CD4 T cells differ in expression of effector molecules. a–d** Single-cell RNA-Sequencing of Nur77-GFP^LO and Nur77-GFP^HI CD4 T cells (2,664 and 1386, respectively) sorted from Nur77-GFP lungs at four weeks post-infection. **a** Data show paired cells from an individual mouse, representative of 4 biological replicates. Colors indicate transcriptomic clusters with manual annotation based on differential expression of marker genes. **b** Feature plots show regions of elevated expression of indicated genes in Nur77-GFP^LO and Nur77-GFP^HI cell populations. For IV label, CD45.2 CITE-seq antibody was injected immediately before harvest. **c** Dot plots show relative abundance (color scale) and prevalence (circle size) of indicated gene expression in each cluster, in Nur77-GFP^LO and Nur77-GFP^HI cell populations. Source data for **a–c** are located in Supplementary Data S3

**d** Columns indicate percent of Nur77-GFP^LO and Nur77-GFP^HI CD4 T cells expressing each gene, paired from 4 individual mice. Asterisks indicate *q* value < 0.05 for paired two-tailed *t* tests adjusted for multiple comparisons, with precise *q* value listed in Supplementary Data S4. **e** Flow cytometry of OX40 surface protein expression on Nur77-GFP^LO and Nur77-GFP^HI CD4 T cells from Nur77-GFP lungs harvested at four weeks post-infection. CD45.2 fluorescent antibody injected immediately before harvest identifies vascular cells. Tetramer^+ cells represent either Ag85B or ESAT-6 specificity. Gated on IV^- Tetramer^+ or bulk live CD4^+ CD44^+ Nur77-GFP^LO or Nur77-GFP^HI cells. *P* values calculated with a paired two-tailed *t*-test, 5–6 mice per group. Error bars indicate standard deviation.

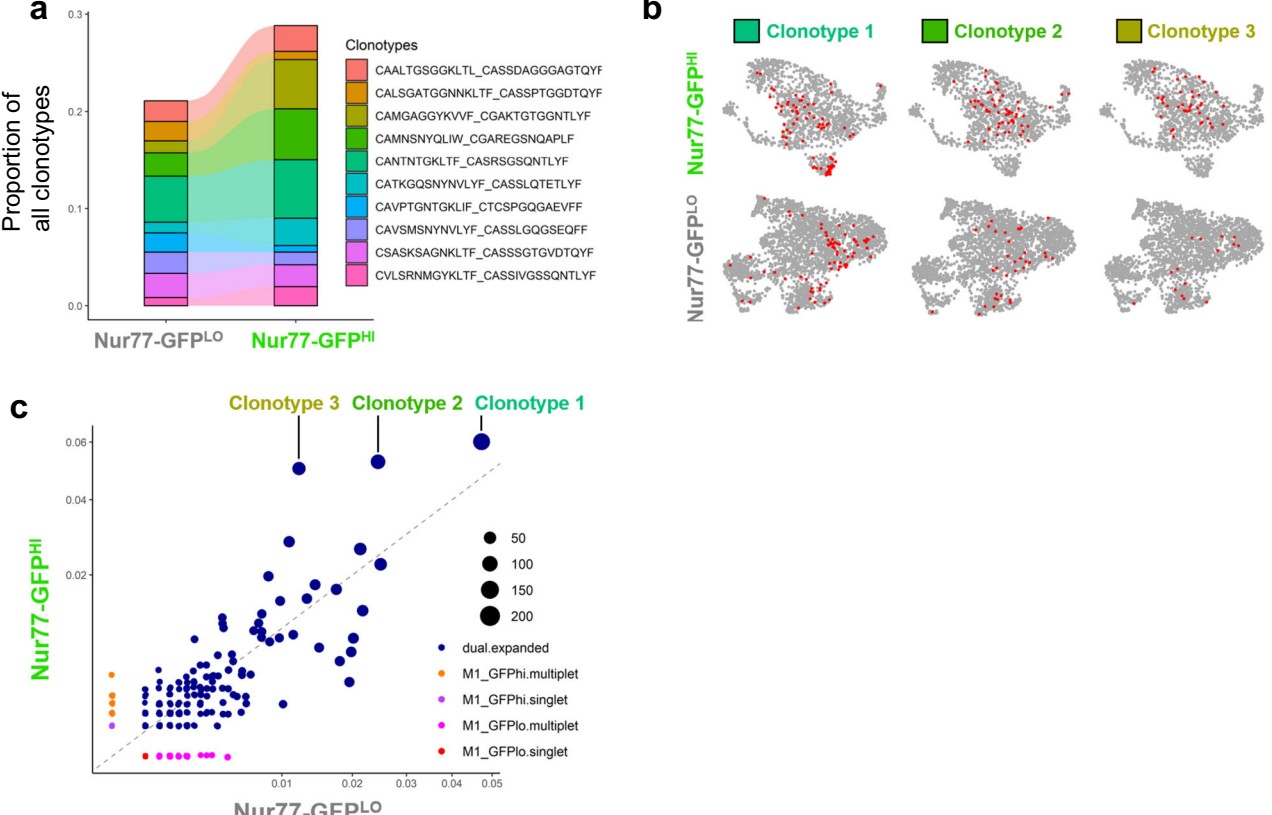

**Fig. 4 | Nur77-GFP^HI and Nur77-GFP^LO CD4 T cells differ in expression of TCR clonotypes. a** Relative abundance of top 10 clonotypes from single-cell TCR-Sequencing of Nur77-GFP^LO and Nur77-GFP^HI CD4 T cells sorted from the lungs of one Nur77-GFP mouse at four weeks post-infection. Samples sorted on live CD3+CD4+CD44+ Nur77-GFP^LO or Nur77-GFP^HI cells. Amino acid sequences represent TCR CDR3 regions expressed as "TCRα_TCRβ". **b** Feature plots show the localization of three expanded TCR clonotypes on gene expression UMAPs from Nur77-GFP^LO and Nur77-GFP^HI CD4 T cells. **c** Scatter plot shows the prevalence of TCR clonotypes among Nur77-GFP^LO vs. Nur77-GFP^HI cells. Blue dots indicate expanded clonotypes. Circle size indicates total abundance of each expanded clonotype.

characterized by extensive regions of pulmonary edema (eosinophilic fluid in airspaces) in contrast to the rare and focal edema in PD-1 blockade and isotype control-treated mice (Fig. S4c and d). We also found that, across treatment groups, acid-fast bacteria (AFB) were not seen in all macrophage aggregates and the number of AF+ bacteria within individual macrophages was highly variable (Fig. S4e). Visual analysis suggested that both isotype control and PD-1 blockade-treated mice had discrete AF+ macrophage aggregates with high densities of bacteria while OX40 agonist-treated mice had more scattered and less discrete AF+ aggregates (Fig. S4e). However, measurement of the largest AF+ macrophage aggregates (n = 5) found no significant difference between treatment groups (Fig. S4f). In conclusion, OX40 agonism-treated lungs have increased cellular inflammation characterized by coalescing macrophage and lymphoid aggregates of greater average size, as well as increased pulmonary edema suggesting more robust lung inflammation.

### OX40 agonism expands CD4 T cell populations in the lung

OX40 stimulation has been demonstrated to potentiate TCR signals in other contexts[31–33], and incorporation of OX40 signaling domains in chimeric antigen receptor-T cell therapies is used to improve cell survival, proliferation, and function[34]. We therefore hypothesized that OX40 agonism improved TB outcomes by expanding the lung CD4 T cell population size and phenotype. We found that OX40 agonist treatment significantly increased the number of polyclonal and Mtb-specific lung CD4 T cells, but not CD8 T cells (Fig. 5d and e). In contrast, PD-1 blockade did not affect the population size of either cell type (Fig. 5c and d). With OX40 agonism, we observed an increase in the

total number of proliferating CD4 T cells but no significant change in the frequency of either CD4 T cells expressing proliferation (Ki-67+) or apoptotic (Caspase-3/7+) markers (Fig. S5a and b). These data suggested that other changes in T cells, such as cell phenotype and localization, may predominantly underlie the beneficial effect of OX40 agonism.

### OX40 agonism induces lasting phenotypic effects on CD4 T cells

To determine how OX40 agonism persistently affects the *Mtb* infection after the cessation of antibody treatment, we investigated the lasting effects on bacterial burden and CD4 T cells at one-month post-treatment (Fig. 6a). We found that the reduction of bacterial burden and increase in CD4 T cell number observed immediately after two weeks of OX40 agonism began to normalize at 42 days post-antibody treatment (Fig. 6b and c). Interestingly, we also observed a change in the localization and phenotype of CD4 T cells after OX40 agonism, with this treatment driving an increase in the percentage of cells localizing to the lung parenchyma (Fig. 6d and e) and a decrease in the fraction of bulk polyclonal and Mtb-tetramer binding cells expressing the terminal differentiation markers KLRG1 and CX3CR1 (Figs. 6f, g, S5d). Although the change in CD4 T cell parenchymal localization was transient, the change in phenotype was persistent, with fewer CD4 T cells in OX40-treated mice expressing KLRG1 and CX3CR1 at one-month post-treatment (Fig. 6g). In addition to these phenotypic changes, we also observed that the percentage of lung CD4 T cells expressing FOXP3 slightly increased immediately post-treatment, (Fig. S5c). *Foxp3* transcript was also specifically and highly expressed in Nur77-GFP^HI CD4 T cell bulk RNA-seq (Fig. 2a) and scRNA-seq (Fig. 3b)

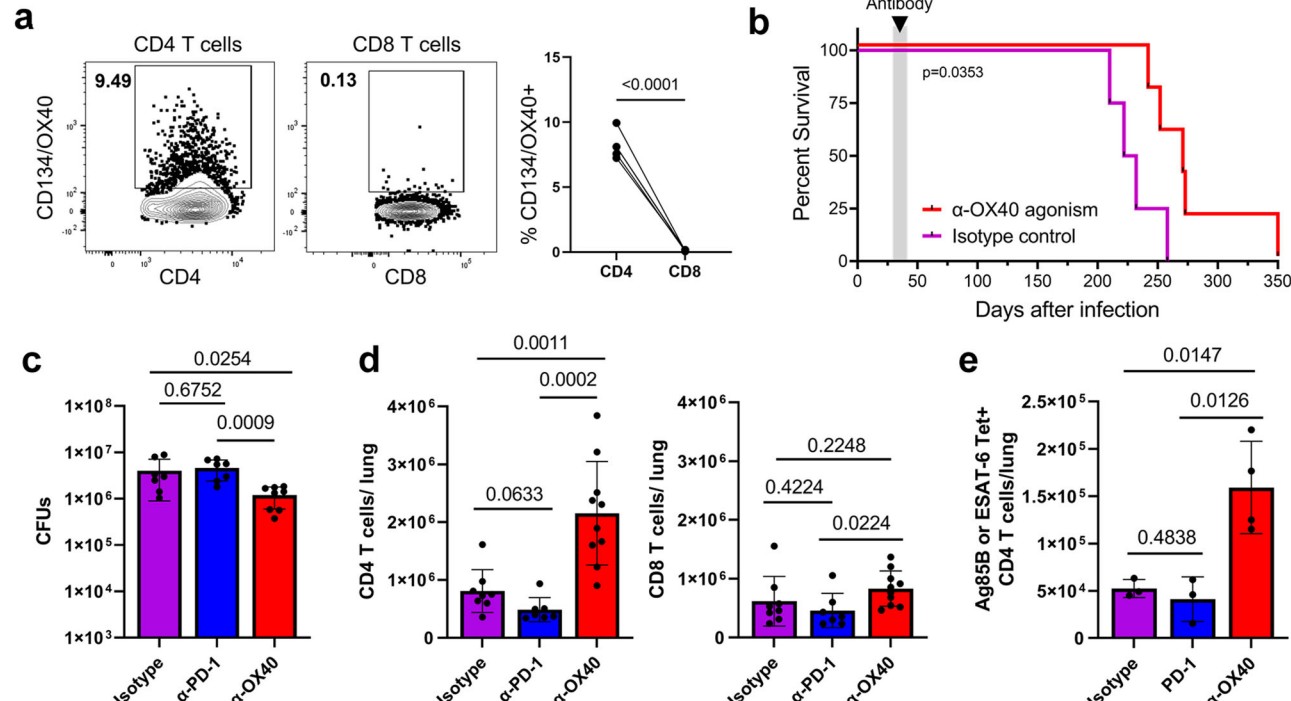

**Fig. 5 | OX40 agonism improves TB outcomes and increases CD4 T cell numbers. a** Flow cytometry of CD4 and CD8 T cells from wildtype C57BL/6 lungs at four weeks post-infection. Gated on live CD4$^{+c}$CD44$^{+}$ or CD8$^{+c}$CD44$^{+}$ cells. *P* value = 0.00007 calculated with a paired two-tailed *t*-test, 4 mice per group. **b–e** Wildtype C57BL/6 mice at four weeks post-infection treated with 100 μg α-OX40 agonist, α-PD-1 blockade, or isotype control antibody injections twice weekly for two weeks. Mice followed for survival or lungs harvested one day after the last antibody treatment for flow cytometry and lung bacterial load. **b** Curves show survival of α-OX40 or isotype control antibody-treated wildtype or TCRα-/- mice. *P* value calculated with a two sided Mantel-Cox test, 4-5 mice per group. **c** Lung bacterial load of α-OX40, α-PD-1, or isotype control antibody-treated mice. One-way ANOVA *p* = 0.0078; *p* values shown in figure calculated with post hoc testing adjusted for multiple comparisons. Mice per group: 6 for isotype, 7 for PD-1, and 8 for OX40.

**d** Number of CD4 and CD8 T cells in the lungs of α-OX40, α-PD-1, or isotype control antibody treated mice. Gated on live CD4$^{+}$ CD44$^{+}$ or CD8$^{+}$ CD44$^{+}$ cells. Cell numbers assessed using counting beads. Data representative of two repeat experiments. One-way ANOVA for CD4 T cells *p* = 0.00002, CD8 T cells *p* = 0.1023; *p* values shown in figure calculated with post hoc testing adjusted for multiple comparisons. *P* value for PD-1 vs OX40 group = 0.00005. Mice per group: 8 for isotype, 7 for PD-1, and 10 for OX40. **e** Number of Tetramer$^{+}$ CD4 T cells in the lungs of α-OX40, α-PD-1, or isotype control antibody-treated mice. Gated on live CD4$^{+}$ CD44$^{+}$ Tetramer$^{+}$ cells. Tetramer$^{+}$ cells represent either Ag85B or ESAT-6 specificity. Cell numbers assessed using counting beads. One-way ANOVA *p* = 0.0047, *p* values shown in figure adjusted for multiple comparisons. Mice per group: 3 for isotype, 3 for PD-1, and Mice per group: 8 for isotype, 7 for PD-1, and 10 for OX40. 4 for OX40. Error bars indicate standard deviation.

datasets, making up approximately 5% of the total cells from this population. These data suggest that OX40 agonism causes both transient and lasting shifts in CD4 T cell localization and phenotype that may underlie that the increased survival of mice undergoing this treatment.

### OX40 agonism offers additive benefit to antibiotic treatment

The future potential of immunotherapies in TB treatment could include the combination of therapeutic antibodies with current TB antibiotics, so we next tested whether there is a benefit of adding OX40 agonist treatments to antibiotic treatment. We treated *Mtb*-infected mice beginning at day 28 post-infection; OX40 agonist antibodies were injected twice weekly for two weeks and antibiotics were given continuously in water until harvest (Fig. 7a). We observed that OX40 agonism significantly reduced bacterial load immediately after the antibody treatment not only as an individual treatment, but also in combination with antibiotics (Fig. 7b). This suggests that OX40 agonism may have beneficial additive effects to current TB antibiotic treatment regimens.

To determine the lasting impact of short-term host-directed immunotherapy with OX40 agonist in combination with antibiotics, we studied later timepoints after antibody treatment had ended but antibiotic treatment continued. We observed that the bacterial load reduction caused by OX40 agonism and antibiotic combination treatment lasted until at least 42 days post-antibody treatment but

waned by 76 days post-treatment (Fig. 7c), suggesting that early OX40 agonism can cause long-term effects when given alongside antibiotics. In contrast, the increased CD4 T cell number caused by OX40 agonism and antibiotic combination treatment persisted, lasting until at two months post-treatment (Fig. 7d). Furthermore, OX40 agonism resulted in a robust change in CD4 T cell phenotype, both without and with combination antibiotics, reducing the percentage of terminally differentiated, CX3CR1/KLRG1-expressing cells at 42 days after the initiation of treatment (Fig. 7e). The predominance of CX3CR1-, KLRG1-CD44+ effector CD4 T cells at this late post-treatment time point is suggestive of a more functional T cell population persisting during antibiotic therapy, which may have important implications for adjunctive host-directed therapies and warrants further study.

### Human CD4 T cells express OX40 at the site of TB infection

To determine whether OX40 expression among T cells at the site of TB infection is conserved across species, we isolated cells from the cerebrospinal fluid (CSF) of human participants diagnosed with HIV-associated TB meningitis. From a single individual with CSF culture positive for *Mtb*, we performed single-cell RNA-Sequencing on live sorted CD4$^{+}$ and CD8$^{+}$ T cells. CD4 T cells formed 4 transcriptional subclusters (Fig. 8a). The dominant CD4 T cell cluster was characterized by evidence of TCR stimulus, expressing the transcription factors *FOS* and *JUNB*, as well as *TNFRSF4* and other costimulatory markers (Fig. 8b and c). One subdominant CD4 cluster showed a cytotoxic phenotype with

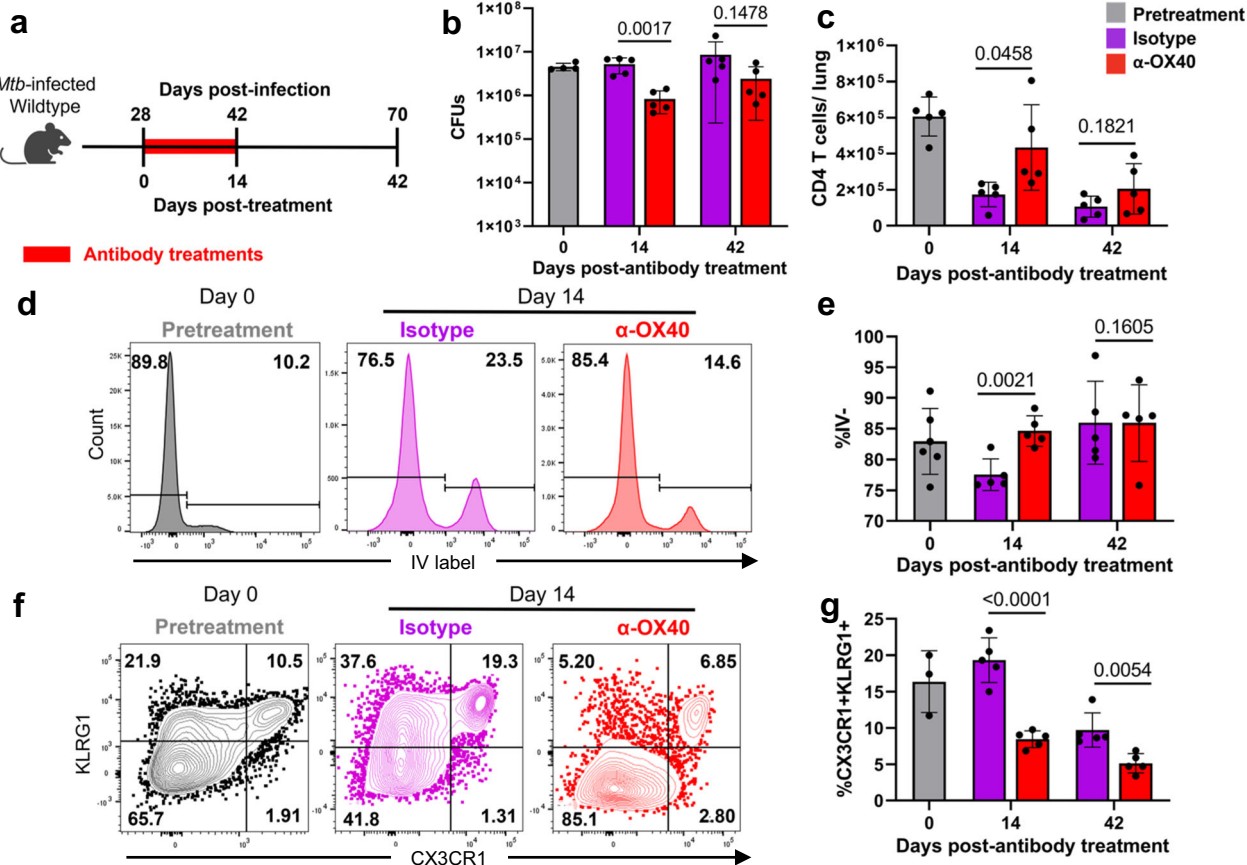

**Fig. 6 | OX40 agonism alters CD4 T cell phenotype and retention at late timepoints. a–g** Wildtype C57BL/6 mice at four weeks post-infection treated with 100 μg α-OX40 agonist or isotype control antibody injections twice weekly for two weeks. Mice harvested at 0, 14, and 14 days post-antibody treatment for flow cytometry or lung bacterial load. **b** Lung bacterial load of α-OX40 or isotype control antibody treated mice. *P* values calculated with an unpaired two tailed *t*-test, adjusted for multiple comparisons. Mice per group: 4 for day 0, 5 each for Isotype or OX40 at day 14, and 5 each for Isotype or OX40 at day 42. **c** Number of CD4 T cells in the lungs of α-OX40 or isotype control antibody treated mice. Gated on live CD4⁺ CD44⁺ cells. Cell numbers assessed using counting beads. *P* values calculated with an unpaired two-tailed *t*-test, adjusted for multiple comparisons, 5 mice per group. **d, e** Frequency of parenchymal CD4 T cells in the lungs of α-OX40 or isotype control antibody-treated mice. CD45.2 fluorescent antibody injected immediately before harvest identifies vascular cells. Gated on live CD4⁺ CD44⁺ IV⁻ cells. *P* values calculated with an unpaired two-tailed *t*-test, adjusted for multiple comparisons. Mice per group: 6 for day 0, 5 each for Isotype or OX40 at day 14, and 5 each for Isotype or OX40 at day 42. **f, g** Frequency of terminally differentiated CD4 T cells in the lungs of α-OX40 or isotype control antibody-treated mice. Gated on live CD4⁺ CD44⁺ CX3CR1⁺ KLRG1⁺ cells. *P* values calculated with an unpaired two-tailed *t*-test, adjusted for multiple comparisons. Mice per group: 3 for day 0, 5 each for Isotype or OX40 at day 14, and 5 each for Isotype or OX40 at day 42. Error bars indicate standard deviation.

*GZMA* and *CCL5* expression. These two major CD4 clusters closely mirrored phenotypes we observed among *Mtb*-specific T cells from the lungs of infected mice (Fig. 2). A second, subdominant CD4 cluster, also expressing *TNFRSF4* and *FOS* co-expressed both type I interferon-stimulated genes (*MX1*) as well as *FOXP3* (Fig. 8c). We found that the final CD4 T cell cluster had the highest density of transcripts aligning to the HIV viral genome and likely represented cells productively infected with HIV. We observed widespread expression of the antiviral defense mediator *ADAR* across all cell types, with the densest expression localizing to the HIV cluster. These data indicated that phenotypes of infection-site CD4 T cells are conserved across species, with a sub-population expressing *TNFRSF4*, likely reflecting recently activated cells. The interferon signature observed among human CSF T cells may reflect the impact of severe HIV co-infection on cellular states at the infection site.

Finally, we used flow cytometry to assess the surface protein expression of OX40 among CD4 and CD8 T cells in the CSF of individuals treated for HIV-associated TBM. Similar to observations in the mouse, we found that OX40 was predominantly expressed on CD4⁺ T cells, with far less frequent expression on CD8 T cells at the site of infection (Fig. 8d).

## Discussion

Effector CD4 T cells are an essential component of protective immunity against TB. Much is already known about mediators required for their functional benefit in controlling *Mtb* infection, but prior studies have been technically limited in a few important ways. The first is the reliance on relatively low-resolution techniques like flow cytometry that require a priori knowledge of presumed pathways of importance, which can impede the discovery of novel mediators of protection. The second is the inability to identify the sub-population of CD4 T cells that are actively engaged with antigen at the time of cell collection, and which presumably actively contribute to protection. Although prior studies have used either TCR-Tg T cells or MHCII tetramers to focus analyses on the antigen-specific T cell response, these approaches limit the study of CD4 T cell function to a few dominant epitope-specificities, which may or may not be representative of the broader poly-clonal response. This is of importance since CD4 T cell responses targeted against different *Mtb* antigens can have different phenotypic characteristics[6,17]. While restimulation using large peptide pools of *Mtb* epitopes have partially mitigated this issue[35], this approach does not capture the activity of cells in situ. The use of ex vivo cytokine production as a readout to identify *Mtb*-specific cells may also be

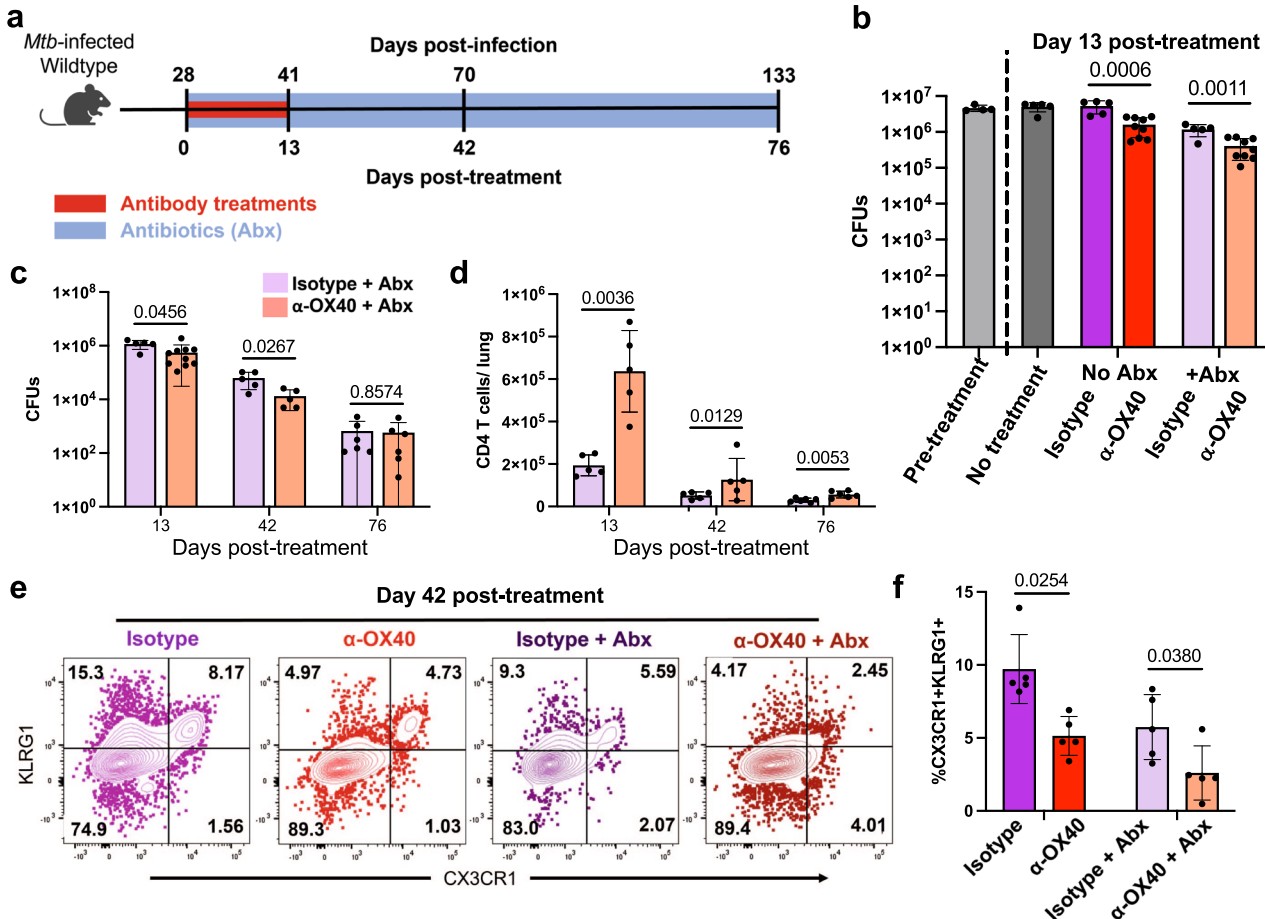

**Fig. 7 | OX40 agonism improves antibiotic treatment outcomes. a–f** Wildtype C57BL/6 mice at four weeks post-infection treated with 100 μg α-OX40 agonist or isotype control antibody injections twice weekly for two weeks. Antibiotic mice also treated from four weeks post-infection to harvest with isoniazid and rifampin in water continuously. Mice harvested at 0, 13, 42, and 76 days post-treatment for flow cytometry of lung bacterial load. **b** Lung bacterial load of α-OX40 antibody or isotype control antibody, with or without antibiotics. Pretreatment wildtype lungs harvested at 0 days post-treatment. Data representative of two repeat experiments. Two-way ANOVA results: antibiotic treatment contributed 43% of variance $p < 0.0001$, antibody treatment contributed 30% of variance $p < 0.0001$, interaction contributed 13% of variance $p = 0.0015$; $p$ values shown in figure calculated with two-tailed $t$ testing adjusted for multiple comparisons. Mice per group: 4 for pre-treatment, 5 for no treatment, 5 for no antibiotics + Isotype, 9 for no antibiotics +

OX40, 5 for antibiotics + Isotype, 9 for antibiotics + OX40. **c, d** CFUs and number of CD4 T cells in the lungs of α-OX40 antibody or isotype control antibody treated mice. Gated on live CD4+ CD44+ cells. Cell numbers assessed using counting beads. $P$ values calculated with an unpaired two tailed $t$-test, adjusted for multiple comparisons. Mice per group: 5 for isotype and 10 for OX40 at day 13, 5 each for isotype and OX40 at day 42, and 6 each for isotype and OX40 at day. **e, f** Frequency of terminally differentiated CD4 T cells in the lungs of α-OX40 antibody or isotype control antibody, with or without antibiotics. Gated on live CD4+ CD44+ CX3CR1+ KLRG1+ cells. Two-way ANOVA results: antibiotic treatment contributed 39% of variance $p = 0.0005$, antibody treatment contributed 27% of variance $p = 0.0005$, interaction contributed 1% of variance $p = 0.43$; $p$ values shown in figure calculated with two-tailed $t$ testing adjusted for multiple comparisons, 5 mice per group. Error bars indicate standard deviation.

biased towards previously known effector phenotypes (e.g. production of IFN-γ, TNF, and IL-2) and requires cell fixation, which prevents downstream applications that require live cell inputs, including functional and transcriptomic analyses. There has been no way to identify the subset of *Mtb* antigen-specific effector CD4 T cells actively responding at the site of infection and learn with an unbiased approach what distinguishes this population phenotypically and functionally.

To address these limitations, we used the Nur77-GFP lymphocyte activation reporter mouse to investigate the effector phenotypes of activated *Mtb*-specific CD4 T cells in the lungs. In addition to traditional flow cytometry, this allowed us to perform live-cell functional assays and both bulk and single-cell RNA sequencing. While other reporter mice exist that are capable of detecting even more recently activated lymphocytes[36], we favored this Nur77-GFP model for its ability to capture cells receiving antigen receptor stimulation within 6-48 hours prior to tissue harvest[12]. We hypothesized that this approach could identify new effector functions, immunotherapy targets, and

cellular phenotypes within the broader CD4 T cell population. Our study adds further dimensions to prior work by defining the transcriptomic phenotypic heterogeneity of *Mtb*-specific CD4 T cells in the lungs. Importantly, we found that the sub-population of activated CD4 T cells is enriched for certain phenotypes, including parenchymal T conventional effectors and T regulatory cells. In line with prior studies, we found that *Foxp3*+ cells in the periphery were commonly Nur77-GFP[HI37], likely reflecting both high-affinity epitope-TCR interactions that characterize Tregs[12], as well as the direct transcriptional relationship of *Nr4a1* and *Foxp3*[38]. Although use of this particular lymphocyte activation reporter ensures a more comprehensive capture of the activated population, one potential caveat is that it may also include strongly autoreactive cells. Whether these activated Tregs are specific for self or *Mtb* antigens, is an important topic for future study. In contrast, we observed that cells expressing a transcriptional profile of terminal differentiation and cytotoxicity were more likely to localize to the vasculature and more likely to be Nur77-GFP[LO] at the time of harvest. Using CITE-Seq with an oligonucleotide-labeled antibody

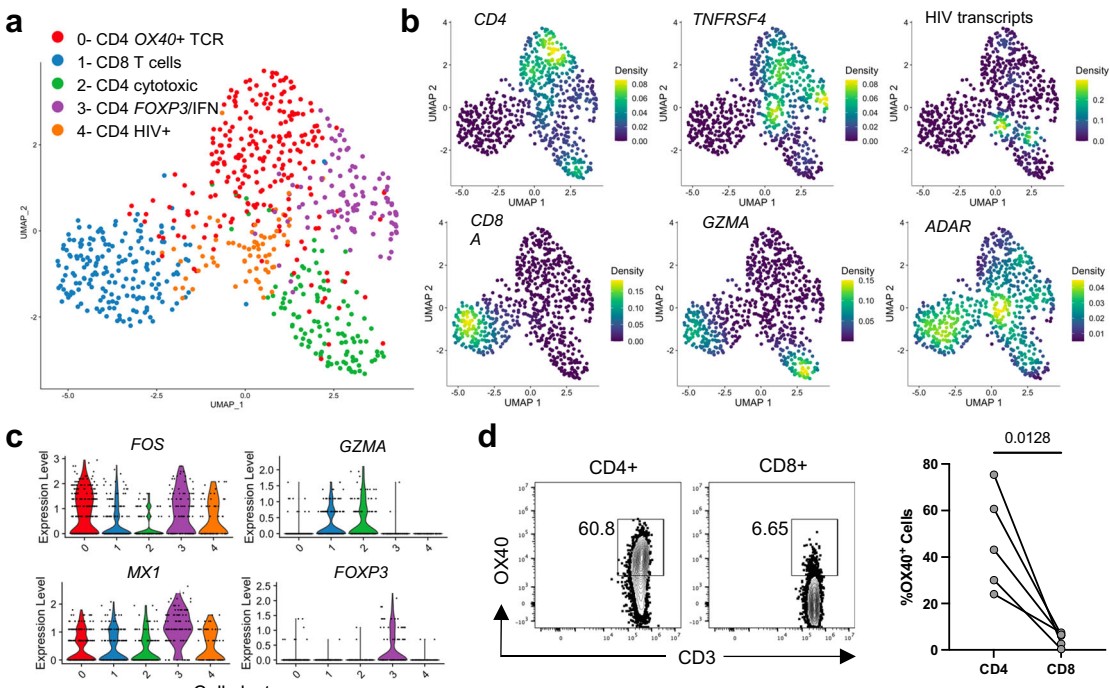

**Fig. 8 | Human CD4 T cells at the site of HIV-associated TB meningitis express OX40. a** UMAP representing single-cell RNA-Sequencing of CD4⁺ and CD8⁺ T cells sorted from cerebrospinal fluid (CSF) cells from a human person with HIV-associated TB meningitis. Sorted cells were gated on live, CD3⁺, CD4⁺ and CD8⁺ cells. Clusters were manually identified according to the differential expression of key genes, indicated as follows. **b** Plots show the relative density of cells expressing each marker gene. **c** Violin plots show each gene's relative expression level by cells assigned to each phenotypic cluster. Source data for **a–c** are located in Supplementary Data S5. **d** OX40 surface expression in cells from the CSF of 5 individuals treated for HIV-associated TB meningitis. Flow cytometry dot plots indicate the percentage of CD4⁺ or CD8⁺ T cells from a single individual that express OX40, while column dot plots show the paired percentage of CD4 or CD8 T cells from 4 individuals. *P* value calculated with a paired two-tailed *t*-test.

injected intravenously immediately prior to the harvest, we identified genes strongly associated with this vascular CD4 T cell population, including the transcription factor *Zeb2* and the trafficking receptor *S1pr5*, both shown by others to contribute to vascular localization of tissue-resident memory T cells[24]. Prior to this study, others had demonstrated this vascular population in the *Mtb*-infected mouse lung, which is enriched for cells expressing CX3CR1 and KLRG1, as we have observed here[8]. Our approach offers further insight into the mechanism that likely underlies the localization of this population, which is not commonly activated, presumably due to its spatial isolation from *Mtb*-infected antigen-presenting cells in the lung parenchyma.

A key characteristic of the Nur77-GFP^HI-activated CD4 T cell population is the expression of multiple costimulatory and coinhibitory markers. This finding validated our approach because expression of many of these markers is induced shortly after antigen receptor stimulation, leading to their use as "Activation Induced Markers" in ex vivo restimulation assays[39]. Costimulatory markers differentially expressed by Nur77-GFP^HI activated cells included several already known to participate in the immune response to *Mtb*, including *Tnfrsf18* (CD153) and *Pdcd1* (PD-1), both of which are required for optimal protection against *Mtb*[13,40]. We also found that *Tnfrsf4* (CD134/OX40) was among the most differentially expressed by activated cells in our flow cytometry, bulk, and single-cell transcriptional analyses. While we did not observe exact concordance of expression of *Nr4a1*/Nur77-GFP and *Tnfrsf4*/OX40, potentially reflecting differences in their temporal expression pattern, we consistently observed that recently activated cells were enriched for cells expressing *Tnfrsf4*/OX40. These repeated results identified *Tnfrsf4*/OX40 as a potentially useful marker gene for activated, parenchymal effector CD4 T cells at the site of infection.

Interestingly, key functional mediators like *Ifng* and *Tnf* were not differentially expressed by activated cells at the transcriptional level; perhaps a consequence of post-transcriptional regulation of these cytokines. Nur77-GFP^HI cells did express more *Il21* than Nur77-GFP^LO cells, consistent with the requirement of this cytokine for optimal protection in the mouse model of TB[23] and suggesting an important role for IL-21 signaling at the site of infection. One important technical limitation of our study is that protein level validation of cytokine expression differences was not possible, because measurement of cytokine production requires ex vivo restimulation.

We found several lines of evidence suggesting that antigen specificity is an important determinant of CD4 T cell activation at the site of infection. We observed that the activated population was enriched for Ag85B and ESAT-6-specific cells, and that expanded TCR clonotypes were relatively more abundant among the Nur77-GFP^HI TCR repertoire. Among three dominant T cell clones, we identified two that were restricted in the Nur77-GFP^HI population to parenchymal T conventional cell clusters, rather than the vasculature. Future studies to identify the antigen specificities of these TCRs could help illuminate which antigens are more likely to elicit vaccine-induced protective, lung-homing CD4 T cell responses. One limitation of the present study is our focus on a single, important time point in the infection, when adaptive immunity curtails *Mtb* growth in the lungs. More work is needed to investigate the plasticity of CD4 T cell phenotypes in relation to activation states over the course of *Mtb* infection.

Although little is known about the role of OX40 in TB immunity, it is an important target for immunotherapy against cancer. Agonism of costimulatory receptors like OX40, 4-1BB, and GITR have shown success in several cancer studies[41–43] and are an emerging alternative to blockade of inhibitory receptors like PD-1. Importantly, the mechanism of action of OX40 agonism may differ from checkpoint blockade,

shifting T cell phenotype and localization, rather than increasing cytokine production. In mice with TB, treatment with a recombinant OX40 ligand enhanced the protective effects of BCG vaccination[44,45], and OX40 deficient mice had a modest though statistically significant reduction in survival after *Mtb* infection[40]. Prior human studies have used OX40 expression after ex vivo restimulation to identify *Mtb*-specific T cells[46], but none have evaluated the expression of this marker by T cells at the site of TB infection. Importantly, we found that at the site of human TB infection, OX40 expression was common among CD4 T cells. *TNFRSF4* was also highly expressed by CD4 T cells in the granulomas of non-human primates infected with *Mtb*[47] and was among a module of genes differentially expressed by airway CD4 T cells responding to high dose IV BCG vaccination, associated with sterilizing protection against subsequent Mtb challenge[48]. We therefore propose that this marker may be a useful way of identifying activated *Mtb* antigen-specific cells at TB infection sites across species. Surprisingly, we found that few CD8 T cells in the *Mtb-infected* mouse lung or human CSF express OX40, in contrast to other infections[26–30]. This finding may suggest that CD8 T cell activation at the site of *Mtb* infection relies less on antigen receptor stimulation, and more on cytokine-driven responses[7,49]. While few CD8 T cells expressed OX40, we could not completely rule out whether CD8 T cells have some effect on improving TB outcomes after OX40 agonism as our survival studies examined completely T cell-deficient TCRα[-/-] mice.

Our study of human CSF T cells in HIV-associated TB meningitis helps to corroborate both the single-cell transcriptional T cell phenotypes observed in the mouse lung, as well as the expression of *TNRSF4*/OX40 among a sub-population of CD4 T cells at the site of infection. It is not yet clear how directly comparable cells from the site of human extra-pulmonary TB infection are to cells from the lungs of mice infected with pulmonary *Mtb* infection. However, the similarity of the transcriptional clusters we observed across species, and the rare opportunity to study human cells from the site of infection while detecting cells expressing high levels of HIV transcripts demonstrates a technical benefit of single cell RNA-sequencing approaches in human TB immunology. We also observed some differences: notably that *FOXP3* expression and type I IFN responsive transcriptional signatures may be a hallmark of effector T cells in HIV-TB coinfection. Future studies should aim to determine whether these unique findings represent common features of CD4 T cells at TB disease sites in humans and whether *TNFRSF4*-expressing CD4 T cells featuring these transcriptional profiles contribute unique roles in HIV-associated and extrapulmonary TB disease.

Based on its preliminary success as a preclinical cancer immunotherapy target, we investigated whether OX40 agonism also had beneficial effects in the mouse model of TB. We found that immunotherapy targeting OX40 + T cells was well-tolerated by *Mtb*-infected mice and resulted in improved survival, reduced lung bacterial load, and increased CD4 T cell numbers. OX40 agonism also caused a marked change in lung inflammation, shifting granuloma structure and increasing both average lesion size and surrounding areas of pulmonary edema. These data suggest that OX40 agonism has a fundamentally different effect on *Mtb*-specific T cell function than PD-1 blockade, which is detrimental due to the dysregulation of IFN-γ production[14–16,50]. We observed that OX40 agonism increased CD4 T cell numbers in the lung; however, the percentage of Ki67 + CD4 T cells was not significantly impacted, suggesting that other changes may underlie its protective benefit. OX40 agonism also transiently increased the fraction of parenchymal CD4 T cells, and significantly shifted CD4 T cell phenotypes towards those expressing FOXP3 and away from terminal differentiation (CX3CR1, KLRG1), suggesting that the lack of adverse effects may be due to expansion of both effector and regulatory cell populations which can maintain inflammatory balance during infection. It will be important to determine whether the beneficial effects of short-term OX40 agonist immunotherapy during early infection can be improved upon with extended treatments during chronic infection, or on the contrary, if repeated stimulation is detrimental, potentially by driving T cell exhaustion or immunopathology. Future studies should tease out other potential effects of OX40 agonism on CD4 T cells, including improved cell survival, cytokine production, and memory cell generation, to discover potential protective biomarkers, as well as to understand the lasting impact of OX40 agonism on histopathology and T cell function that confers enhanced long-term survival despite persistent chronic infection. Our findings support the need to further investigate OX40 agonism and other T cell-targeted immunotherapies as adjuncts to antibiotic treatment for human TB. This dual approach to TB therapy could allow the targeted expansion of functional CD4 T cells with immunotherapy despite the rapid reduction of antigenic bacterial load that occurs with antibiotics, potentially leading to more efficient treatment regimens for TB and enhancing protection against reinfection.

## Methods

### Mouse aerosol infection

Prior to infection, mice were housed in a specific pathogen-free facility. Control animals for infections were co-housed in the same facility. Mice in equal numbers from both male and female sexes were infected at 8-16 weeks of age with a standard dose of aerosolized *Mycobacterium tuberculosis* H37Rv, targeting 100-200 CFUs, using an Inhalation Exposure Unit (Glas-Col). Nur77-GFP (strain 016617) and *TCRα[-/-]* (strain 002116) mice were initially purchased from Jackson Labs, then bred in house. Following infection, mice in both experimental groups and controls were housed in the same ABSL3-specific pathogen-free facility. Inoculum size for each infection was confirmed with harvest of 3 C57BL/6 J (Jackson Labs strain 000664) mice 24 hours post-infection. Lungs were homogenized in PBS with Tween 80, then plated on 7H10 agar plates and incubated at 37 °C for 3-4 weeks prior to counting colony-forming units. Schematic diagrams in Figs. 1, 6, and 7 representing mouse experimental infections were partially created with BioRender.com.

### Lung tissue harvest for flow cytometry and CFU plating

For experiments involving in vivo peptide stimulation, mice were injected via the tail vein with 100 μg synthetic Ag85B[240-254] (FQDAY-NAAGGHNAVF) peptide 6 hours before euthanasia. For identification of vascular cells during flow cytometry, mice were injected via the tail vein with 3 μg CD45 rF710 (Tonbo Biosciences) immediately before euthanasia. Mice were euthanized with $CO_2$ and lungs were suspended in a digestion buffer containing DMEM, FBS, Collagenase D, DNase, Heparin, $CaCl_2$, and $MgCl_2$. Lungs were incubated at 37 °C for 30 minutes and dissociated with a gentleMACS dissociator (Miltenyi Biotec 130-093-235), according to the manufacturer's protocol. To isolate cells, lung suspensions were filtered through a 70 μm cell strainer. For experiments with CFU plating, a lung suspension aliquot was taken after cell filtration, serially diluted in PBS with Tween 80, and plated on 7H10 agar plates. Plates were incubated at 37 °C for 3-4 weeks then CFUs were calculated by counting single colonies. For experiments with flow cytometry, red blood cells were removed by incubating cells in ACK lysis buffer for 5 minutes. For tetramer staining, cells were incubated in 50 nM dasatinib (Sigma-Aldrich SML2589) for 15 minutes, tetramer (NIH Tetramer Core Facility) was added at 1:100 dilution, then cells were incubated for one hour at room temperature or 37 C. For surface marker staining, cells were incubated with fluorescently labeled antibodies at 4 °C for 30 minutes. Antibodies used for flow cytometry are found in Supplementary Table S1. Precision count beads (BioLegend 424902) were added immediately before flow cytometry to enable cell enumeration. Flow cytometry was performed with a BD Fortessa H1770 (BD Biosciences) or LSR Fortessa X-20

(BD Biosciences) at the University of Minnesota Flow Cytometry Resource. Data analysis was performed using FlowJo software.

## Sorting for adoptive transfer and RNA sequencing

Sorting for the tetramer-enriched scRNA-Seq was performed using a CD4$^+$ T cell isolation kit (Miltenyi Biotec 130-104-454) followed by tetramer staining and enrichment of tetramer-specific cells with anti-PE microbeads (Miltenyi Biotec 130-048-801). All other scRNA-Seq and adoptive transfer experiments used fluorescence-activated cell sorting. Cells from lung suspensions were prepared and stained for flow cytometry as above, then sorted on a MA900 Multi-Application Cell Sorter (Sony Biotechnology) into DMEM with 40% FBS for post-sort adoptive transfer or scRNA-Seq, or into TRI Reagent (Sigma-Aldrich T9424) for RNA sequencing. All sorted samples were pre-gated on CD3$^+$, CD4$^+$, CD8$^-$, CD44$^+$ live, single-cells with the exception of bulk CD4 T cells magnetically sorted for adoptive transfer which were not stained with CD3. To isolate recently activated and not recently activated CD4 T cell populations from Nur77-GFP reporter mice, samples were also sorted based on the highest and lowest ~30% of GFP fluorescence intensity. Cells for adoptive transfer were injected via the tail vein into recipient mice. Adoptive transfer to determine Nur77-GFP$^{LO}$ and Nur77-GFP$^{HI}$ CD4 T cell protection included 2-5 × 10$^5$ cells per mouse. Sorting of Nur77-GFP$^{LO}$ and Nur77-GFP$^{HI}$ CD4 T cells for RNA sequencing yielded 1–2 × 10$^5$ cells per sample.

## Human cerebrospinal fluid cells

Cells from the CSF of individuals treated for HIV-associated TB meningitis were obtained from participants who provided informed consent for CSF specimen collection for research purposes (UMN IRB: 0308M51329). Cells were obtained on either the day of study enrollment prior to treatment or on day 1 of TBM therapy. CSF samples were initially stored at 4 °C and cells were isolated by centrifugation at 300x $g$ for 10 minutes within 6 hours of collection. Cell pellets for flow cytometry analysis were resuspended in TransFix fixative (Cytomark TFB-01-1) for 24-72 hours, as per manufacturer's instructions, then stained with antibodies prior to analysis on a Cytoflex flow cytometer. Antibodies and other reagents used for flow cytometry are found in Supplementary Table S1. The CSF cell pellet used for scRNA-Seq was resuspended in freezing medium (FCS with 10% DMSO) prior to cryopreservation at −80 °C using a Mr. Frosty and final storage at −150 °C. Within 6 months of cryopreservation, cells were transported at −50 °C, cryo-recovered in batch, washed with complete RPMI medium, and labeled with a fluorescent viability dye and antibodies for sorting on a MA900 Multi-Application Cell Sorter (Sony Biotechnology) into DMEM with 40% FBS. All sorted samples were pre-gated on CD3$^+$, CD4$^+$ or CD8$^+$, live, single-cells. 5,000 sorted cells from each CD4$^+$ and CD8$^+$ population were pooled and used for scRNA-Seq analysis as below.

## RNA extraction and bulk RNA-Seq

RNA extraction was used to prepare sorted cells suspended in TRI Reagent for submission for RNA sequencing. Samples in TRI Reagent were centrifuged at 1000 x $g$ for 10 minutes. Samples were resuspended in chloroform and incubated for 3 minutes then centrifuged at 12,000x $g$ for 15 minutes. The supernatant was collected, isopropyl alcohol added, incubated for 10 minutes, then centrifuged at 7500x $g$ for 5 minutes. The supernatant was discarded then samples were air dried before resuspension in RNase-free water. RNA was submitted to GeneWiz from Azenta Life Sciences for RNA sequencing and differential gene expression analysis using DeSeq2. Data visualizations were prepared using EnhancedVolcano and pheatmap for scaled gene expression analysis, using R version 1.4.

## Single-cell RNA-sequencing

Cells sorted into DMEM with 40% FBS were used for scRNA-Seq sample preparation. After sorting, cell number was confirmed with the Luna II

(Logos Biosystems). The appropriate number of cells necessary to achieve a target cell input of 2-10,000 cells was resuspended and used to generate GEMs (Gel Bead-In Emulsion). The Chromium Next GEM Single-cell 5′ Gel beads v2 kit (10X Genomics) was used and manufacturer instructions were followed for GEM generation through cDNA synthesis. A Chromium Control was used for GEM generation and barcoding then a thermocycler was used for cDNA synthesis. cDNA was submitted to the University of Minnesota Genomics Center for scRNA-Seq. Tetramer-enriched scRNA-seq data pooled from three biological replicates. Nur77-GFP$^{HI}$ and Nur77-GFP$^{LO}$ scRNA-seq data representative of four biological replicates.

## Mouse single-cell RNA-sequencing data analysis

Mouse scRNA-Seq raw expression data was mapped with Cell Ranger (version 7.0.0) for alignment to a custom reference comprised of the mouse genome with an additional EGFP reporter gene. The resultant raw count matrix was imported into a R pipeline with Seurat version 4.0.6[51]. Cells were filtered for a minimum of 200 and maximum of 2,500 genes per cell, mitochondrial genes less than 10%, ribosomal genes less than 20%, and expression of Cd3e over 2 counts. The Seurat default methods were used, with SCtransform for data normalization and scaling, selection of variable genes for PCA and clustering analysis, and regression of mitochondrial and ribosomal genes. For single-cell TCR-sequencing analysis, TCR sequences were also regressed during SCtransform, to minimize impact on gene expression clustering. To assess the relationships between cells, cells were clustered with FindClusters using a resolution 0.3 for GFP$^{HI}$ samples and 0.25 for GFP$^{LO}$ samples to remove redundant clustering. To compare the percentage of Nur77-GFP$^{LO}$ and HI T cells expressing a gene of interest, the percent of cells per each sample from four individual mice, with at least one read was calculated and percentages compared using a paired $t$-test with adjustment for multiple hypothesis testing Seurat functions and methods were used for gene expression results visualization. For analysis of TCR sequencing data, clonotype data was merged with the Seurat dataset using a R pipeline with scRepertoire version 1.6.0[52]. scRepertoire functions and methods were used for TCR expression results visualization.

## Human single-cell RNA-aequencing data analysis

Human scRNA-Seq raw expression data were mapped with Cell Ranger (version 7.0.0) for alignment to a custom reference comprised of the human genome with segments of an Ugandan HIV pro-viral genome (GenBank: AB098330.1), relevant to the geographic region from which the specimens were originally isolated[53]. The resultant raw count matrix was imported into an R pipeline with Seurat version 4.0.6. Cells were filtered for a minimum of 200 and maximum of 3,000 genes per cell, mitochondrial genes less than 25%, ribosomal genes less than 40%. The Seurat default methods were used, with SCtransform for data normalization and scaling, selection of variable genes for PCA and clustering analysis, and regression of mitochondrial and ribosomal genes. To assess the relationships between cells, cells were clustered with FindClusters using a resolution of 1.0. Dimensionality and violin plots were generated using Seurat, while density plots were generated with Nebulosa[54].

## Mouse treatments

Mice were treated with OX40 agonist (OX-86, Cat# BE0031), PD-1 blockade (29 F.1A12, Cat# BE0273), and isotype control antibodies (HRPN, Cat# BE0088) obtained from BioXCell, and the antibiotics isoniazid (ThermoFisher 11-101-4575) and rifampin (Sigma-Aldrich R3501) in various experiments. In vivo antibody treatments were performed with intraperitoneal injections of 100 µg per dose every 3-4 days. Antibiotic treatments were performed by adding 0.1 g/L rifampin and 0.1 g/L isoniazid to a water bottle and replaced weekly.

## Histopathology

Mice were euthanized with $CO_2$ and lungs were suspended in 10% formalin at 4 °C for 24 hours. Fixed lungs were submitted to the University of Minnesota Clinical and Translational Science Institute for paraffin embedding, sectioning, staining with hematoxylin and eosin (HE) and acid fast (AF) stains. Stained lungs on glass slides were then given to the Comparative Pathology Laboratory at the University of Iowa for evaluation. Stained tissue sections were evaluated and scored by a boarded comparative veterinary pathologist using a post examination method of masking the observer to group assignment[55,56]. Tissues were ordinally scored for pulmonary edema (eosinophilic fluid in airspaces) and cellular inflammation on extent of lung involvement for each lung. Involvement scores range from: 0, none; 1, <25%; 2, 26-50%; 3, 51-75%; and 4, >75% of lung fields. Inflammatory aggregates were identified, and the minor axis diameter assessed in a masked manner. The number of aggregates, the average diameter in microns and the sum distance (mm) of all foci were assessed as metrics of the severity of cellular inflammation. The maximal size of macrophage aggregates containing AF+ bacteria were evaluated by quantifying the least dimensional diameter (microns) in the five largest aggregates.

## Statistics

For flow cytometry experiments comparing the percentage or total number of cells gated from a population, a two-tailed student's $t$ test was used, assuming a normal distribution among biological replicates, to calculate $p$ values with a threshold for statistical significance of $p < 0.05$. In instances where two populations were compared from the same individual, a paired $t$ test was used. To compare means of three different treatment groups, one-way ANOVA was used, with post hoc unpaired $t$ testing. To compare two treatment groups at each of several time points, unpaired two-tailed $t$ tests were used, with $p$ value adjustment for multiple comparisons. To compare treatment with antibiotics and OX40 agonism, either in isolation or as combination therapy, two-way ANOVA was used to determine the contribution of each treatment to variance in the model and whether an interaction between existed. To compare the effect of OX40 agonism on CFUs or cell surface marker expression with or without antibiotics, unpaired $t$ tests were used with adjustment for multiple comparisons. To compare differences observed in mouse survival after *Mtb* infection, $p$ values were calculated with a Mantel-Cox test, with a threshold for statistical significance of $p < 0.05$. For bulk RNA-Sequencing, DeSeq2 was used to identify genes with a log2 fold change > 1 and -log adjusted $p$ value > 1 as differentially expressed between populations. For single cell RNA-Sequencing analysis comparing the percentage of cells expressing genes of interest among Nur77-GFP[HI] and [LO] cell populations, multiple paired $t$ tests were used, with a threshold for statistical significance, adjusted for multiple hypothesis testing, of $q < 0.05$. For all figures, error bars indicate standard deviation.

## Study approval

Approval for mouse studies was received by the University of Minnesota Institutional Animal Care and Use Committee. Studies involving specimens from human participants were approved by the University of Minnesota Institutional Review Board. Written informed consent for research on these specimens was received prior to participation.

## Reporting summary

Further information on research design is available in the Nature Portfolio Reporting Summary linked to this article.

## Data availability

Source data are provided with this paper. The data that support the findings of this study have been deposited at the Gene Expression Omnibus (GEO) with accession GSE235800. All other data are available in the article and its Supplementary files or from the corresponding author upon request. Source data are provided with this paper.

## Code availability

The code used to process and analyze single cell RNA-Sequencing experiments has been posted at https://github.com/tylerbold/Activated_CD4_T_cells_in_tuberculosis_express_OX40_a_target_for_host_directed_immunotherapy, assigned: https://doi.org/10.5281/zenodo.10076323.

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

## Acknowledgements

We thank the University of Minnesota Biosafety Level 3 laboratory program, Flow Cytometry core facility, and Genomics Center, as well as the

Infectious Diseases Institute, Makerere University, Meningitis Research Team. This work was funded by K08AI150425 (TDB), R01 R01AI173780 (TDB), T32HL007741 (ARG), T32AI055433 (JMT), and R01AI162786 (DRB and DBM).

## Author contributions

A.R.G: Funding acquisition, Investigation, Methodology, Validation, Data curation, Software, Formal Analysis, Visualization, Writing – original draft. C.E.R: Methodology, Data curation, Project administration. T.V.M.: Methodology, Data curation, Software. J.M.T.: Methodology, Data curation, Software. D.K.M.: Methodology, Formal Analysis, Visualization, Writing. K.S.: Methodology. J.S.: Project administration, Investigation, Methodology, Validation, Data curation. S.O.: Project administration, Investigation, Methodology, Validation, Data curation. D.B.M.: Funding acquisition, Project administration, Supervision. F.V.C.: Project administration, Resources, Data curation. D.R.B.: Project administration, Data curation, Funding acquisition. T.D.B.: Conceptualization, Data curation, Formal Analysis, Funding acquisition, Investigation, Methodology, Project administration, Software, Supervision, Validation, Visualization, Writing – original draft, Writing – review & editing.

## Competing interests

The authors declare no competing interests.
