## [Peer Review File · Nature Communications]

Recently activated CD4 T cells in tuberculosis express OX40 as a target for host-directed immunotherapyREVIEWER COMMENTS

Reviewer #1 (Remarks to the Author):

In this manuscript, Gress et al use Nur77-GFP reporter mice to characterize CD4 T cells that have recently seen Ag in the lungs of Mtb infected mice. They find that GFP-high cells are highly enriched in the in the tissue parenchyma compared to GFP-low cells. Moreover GFP-high CD4 T cells are more protective upon transfer into infected recipient mice. The authors use two different transcriptomic approaches, including a scRNAseq analysis, to characterize the unique properties of Nur77-GFP-high cells. The authors find that Nur77-GFP-high cells express high levels of the costimulatory molecule OX-40, so the authors explore the possibility of targeting OX-40 with agonist mAbs as a host directed therapy (HDT) strategy. It is found that cross-linking OX-40 in Mtb infected mice reduces bacterial loads and extends host survival. Lastly, the authors perform scRNAseq analysis of bulk T cells purified from the cerebrospinal fluid of individuals with HIV-TB co-infection associated meningitis and find the expression of OX-40 on CD4 T cells. The authors major conclusions are that CD4 T cells express OX-40 at sites of Mtb infection, and this costimulatory molecule may be a target for TB HDT. This manuscript contains a lot of transcriptomic data on T cells in tuberculosis, which are valuable data sets. There is interest in better understanding the nature of T cells that actively recognize Ag at sites of infection. Moreover, there is interest in identifying potential targets for HDT of TB. There are, however, concerns regarding the novelty of the major conclusions regarding OX-40 as well as concerns with the study design.

Major concerns

1) The two major conclusions are that T cell express OX-40 and that it is a target for HDT of TB. The expression of OX-40 on CD4 T cells during Mtb infection, however, is not novel or unexpected. One of the most novel findings of the manuscript relates to the potential of treating tuberculosis by boosting CD4 T cell responses by co-stimulating OX-40 with cross-linking antibodies, but the impacts of OX-40 agonism were minimal. The difference in bacterial loads after anti-OX-40 is 2-fold at best, the survival experiment in 5B seems to represent a single experiment with n=3 mice, and OX-40 cross linking didn't enhance bacterial control as an adjunctive to antibiotic therapy. This seems to be a relatively weak case for targeting OX-40. Have the authors tried agonizing OX-40 in other more susceptible

mouse models including SP140^{-/-} mice?

2) To compare cells that are Ag-stimulated vs. not stimulated, the authors gated on GFP-high or GFP-low cells. There are several limitations to this approach that could have been improved on. Firstly, GFP-high cells are enriched in the lung parenchyma, so the differences found in this manuscript largely reflected the qualities of intravascular vs parenchymal T cells, which are populations that the authors point out have already been extensively previously described in Mtb infected mice. A much more informative comparison would have been to compare GFP-high vs GFP-low cells among the iv stain negative cells. This way, T cells of similar differentiation states and tissue localization would be compared. This data may be available in the scRNAseq exp shown in Fig 3, but it is not analyzed for this comparison. Secondly, the comparison of bulk GFP⁺ vs GFP⁻ cells is difficult to interpret due to the likely difference in the abundance of Ag-specific T cells in these two populations. I think it is safe to expect that non-specific bystander T cells will be greatly enriched in the reporter negative cells. In one of the studies the authors perform tetramer pull down enrichment prior to sequencing, but the percentage of cells that are Ag-specific after this approach is still expected to be relatively low. Therefore, the pull-down experiment does not sufficiently address the caveat. Thus, the most informative approach to this experiment would have been to first isolate Ag-specific T cells that are in the parenchyma and then compare reporter high vs low. For example, this could be done by FACS purifying tetramer⁺ cells or by crossing the reporter to a ESAT-6 specific TCR Tg mouse.

3) The Nur77 reporter mice provide an obvious opportunity to examine the sites of T cell stimulation in the lungs, but this is not addressed. For example, do Nur77 reporter positive cells localize near bacteria infected cells?

Minor points

1) In Figure 1B, it appears that Mtb-specific T cells express very little GFP. It is hard to tell from this data if ESAT-6 and Ag85b specific T cells are any different than other T cells in the lungs. A simple FACS plot of tetramer vs reporter would help the reader better evaluate the data. Just how robust is this reporter in identifying Ag-stimulated cells?

2) In Figure 1G, why do all the donor cells migrate into the parenchyma after transfer? It

would be expected that some cells would be iv+ after transfer.

3) I appreciate the attempt to confirm their murine model data in human T cells obtained from an infection site in patients. However, it is not clear how CSF cells from TB/HIV meningitis relate to lungs of Mtb infected mice.

4) The data in 1H show a 10-fold reduction in bacterial loads just 1 weeks after transfer. This seems like a much larger effect than expected. This experiment would benefit from a control group did not receive T cells.

5) There is no analysis of GFP+ and GFP- CD8 T cells. Given the increasing interest in the field in the poor ability of CD8 T cells to recognize Mtb infected macrophages, it seems this is a missed opportunity.

6) The gating strategy for Nur77-GFP-high and low are not well defined, as it is not clear why the top 1/3 and bott 1/3 were chosen for downstream analysis when the gating in 1A shows two populations.

7) Figure 5 only shows bulk T cell responses and does not examine the frequency of antigen-specific T cells or functional changes after OX40 treatment.

Reviewer #2 (Remarks to the Author):

In the manuscript titled “Activated CD4 T cells in tuberculosis express OX40, a novel target for host directed immunotherapy” Gress et al. utilize a Nur77-GFP reporter mouse to identify recently activated CD4+ T cells at the site of Mtb infection. This reporter mouse offers a tool to identify activated CD4+ T cells that are antigen specific for Mtb without the limitations of needing to select for epitope specificities. The authors demonstrate that the Nur77-GFP^{hi} T cell subset at 4 weeks post-infection is protective against Mtb infection. They used several methods (bulk RNAseq, scRNAseq, scTCR sequencing) to perform a thorough characterization of this cell population which shows a heterogenous mix of effector T cells with a striking upregulation of the costimulatory molecule OX40. They show how a OX40 agonist treatment can improve disease outcome and relate their findings to human data where OX40+ CD4+ T cells were identified in the cerebrospinal fluid of patients with HIV-associated TB meningitis. This manuscript is the first to describe OX40 as a marker for Mtb-specific T cells and to suggest OX40 agonism alone as a viable immunotherapy for treatment of tuberculosis.

Major concerns:

1. The reporter mouse used in this paper is an effective tool to study recently activated T cells following an infection. However, the authors did not show data to properly validate the model. Fig 1A should include an uninfected and infected wildtype control stained with a FACS Nur77 antibody to show that the transgene does not impact TCR activation at baseline, and also adequately represents the regular numbers of Nur77 T cell activation. Furthermore, the authors use the timepoint of 4 weeks post-infection for the entirety of the manuscript citing this as the timepoint at which T cells begin to control Mtb growth from Mogues et al (2001). However, Mogues et al. report day 20 post-infection as the timepoint where T cell responses reduce bacterial growth. Despite this, the authors should definitively show that within their model system, day 28 is a timepoint at which protective T cell responses are observed, that way, the comprehensive characterization of T cell responses on this day will be proven relevant. Thus, lung CFUs should be shown as a kinetic following infection. In addition to this, due to the transient nature of Nur77, the authors should show Nur77 expression overtime following infection to perhaps indicate a peak T cell activation at the chosen timepoint, in parallel with a reduction or control of bacterial growth.

2. In this study, the authors use a Nur77-GFP reporter mouse to characterize activated vs bystander CD4⁺ T cell following Mtb infection. The Nur77-GFP^{hi} cell subset is described as Mtb-specific activated T cells and shown to have significantly increased effector function and protective capacity. The Nur77-GFP^{lo} and -GFP^{hi} are gated on CD3⁺CD44⁺ T cells, respectively representing activated and bystander subsets. The terminology used to describe the Nur77-GFP^{lo} and -GFP^{hi} cells is misleading since, by definition, CD44⁺ T cells are considered “activated”. The Nur77-GFP^{hi} T cells should rather be described as “recently activated”, considering the transient expression of Nur77 following TCR stimulation. Furthermore, based on the presented gating strategy, CD44⁺ Nur77-GFP^{lo} would include CD4⁺ memory T cells (effector, central and resident) that are present in the lungs and not antigen-specific for Mtb (Baaten et al. 2010). Therefore, the results from the comparison of Nur77-GFP^{hi} and Nur77-GFP^{lo} cells are expected. It is predictable that recently activated antigen-specific effector T cells (Nur77-GFP^{hi}) are more protective against Mtb than a heterogenous subset of memory/effector and antigen-specific/non-specific T cells. Although the experiments shown in Fig 1D-1H are important to validate this aspect of the model, the

authors should highlight the underlying bias that grants Nur77-GFP^{hi} a more protective phenotype rather than presenting the described subset as a distinct and protective cell population.

3. The study shows that a OX40 agonist treatment helps drive CD4 effector T cell responses thereby conferring protection against Mtb infection. Hence, the authors suggest further investigation into the clinical applications of OX40 agonism for treatment of tuberculosis. However, the data fails to show an additive effect of the combined treatment of antibiotics and OX40 agonism which represents the most relevant condition clinically. We suggest repeating this experiment with longer timepoints to ascertain potential long-term bacterial reduction or control. The reviewer also suggests comparing OX40 agonism to PD-1 blockade therapy to demonstrate the benefit of OX40 agonism compared to a widely used immunotherapy. In conjunction, seeing as PD-1 blockade results in various adverse effects (ie. IFN γ dysfunction), the authors could show that OX40 agonism does not have such outcomes and should therefore be considered as a more viable immunotherapy.

Minor concerns:

Methods:

- The standard dose of aerosolized Mtb is 50-100 CFUs. A dose of 100-200 CFUs is used in this study and should therefore not be called a "low dose", but a standard dose or high dose.

Figure 1:

- Fig 1D-1H: There is inconsistency in the experimental setup and readouts of the two adoptive transfers.

o What is the reasoning behind transferring the T cells in the TCR α ^{-/-} recipient mice 1 week post-infection when 4 weeks is the timepoint of transfer into the wildtype recipient and the timepoint at which T cell responses are claimed to be protective during infection.

o Ideally, survival data and CFU data should be shown for both adoptive transfers.

o A no transfer control should be included in the adoptive transfer in wildtype recipients.

o The presence of Nur77-GFP cells in the vascular and parenchymal compartment of the lungs should be shown for the adoptive transfer in TCR α ^{-/-} recipients. The lymphopenic

environment could be impacting the proliferation and localization of transferred T cells.

- Fig 1G: There appears to be no detectable levels of Nur77-GFPlo or Nur77-GFP^{hi} cells in the vasculature of the recipient mice which seems unusual if the T cells were transferred intravenously. Additionally, Nur77-GFPlo cells are described to usually be found in the vascular compartment, which suggests that this adoptive transfer is not adequately recapitulating the true localization of the T cell subsets. Therefore, the conclusions made about the protective capacity of Nur77-GFPlo cells could be incorrect.

Figure 4:

- In the paper that first described this reporter mouse, Moran et al. show that Nur77-GFP expression is also a sensitive reporter for the measure TCR signal strength. Describing the three major clonotypes as the ones who can confer preferential activation could be erroneous since Nur77-GFPlo T cells also show antigen-specificity and were therefore at some point also activated. The authors should describe these findings as the clonotypes that induce strong TCR stimulation, rather than activation.

- This data shows that the clonotypes between Nur77-GFPlo and -GFP^{hi} are similar which indicates that a large portion of Nur77-GFPlo T cells are Mtb-specific. In light of this, the term “bystander” used to define Nur77-GFPlo cells should be revised as bystander T cells are typically defined by activation of inflammatory signals without antigen recognition.

Figure 5:

- Fig 5D, 5E: An isotype control should be included just as in 5C.

- The axes on both CFU graphs should be the same.

- Fig 5D: The sentence in the figure legend “Cell numbers assessed using flow cytometry with counting beads, gated on live CD3+CD44+ CD4+ or CD8+ cells” is misplaced and should follow the 5D description not 5E.

- Fig 5E: The authors specify that that the antibiotic treatment and OX40 agonist treatment both lasted two weeks. However, the pretreatment group and treatment group are 11 days apart, indicating that the treatment regiment could not have lasted two weeks.

Reviewer #3 (Remarks to the Author):

In this manuscript by Gress and colleagues they investigate T cell activation within the lung parenchyma in response to murine TB infection. They employ the established Nur77-GFP model (see comments below), which reports activity of Nr4a1 (an immediate early TCR-activated gene used as a proxy for T cell activation). Infection of mice leads to a large increase in Nur77-GFP^{hi} T cells, i.e. recently strongly TCR-stimulated cells, that they further interrogate in comparison to Nur77-GFP^{lo} counterparts. Nur77-GFP^{hi} T cells are more greatly enriched for antigen-specific T cells and have superior protective capacity compared to Nur77-GFP^{low} T cells. The authors cite this as evidence for a functional difference in the population (see comment below). Nur77GFP^{hi} cells comprise a heterogeneous population of Treg and effector cells, and Nur77GFP^{hi} have many hallmarks of strong TCR signalling. Nur77GFP^{hi} cells have higher levels of co-receptors (such as OX40, GITR, 41BB) and co-inhibitory receptors (Lag3, CTLA-4). Whilst Nur77GFP^{hi} and low cells share similar dominant clones, the Nur77GFP^{hi} fraction shows greater expansion of the top three dominant clonotypes. The authors show that OX40 is a selective marker of Nur77-GFP^{hi} cells, that is largely restricted to CD4 but not CD8 T cells. OX40 agonism enhances survival of TB infected B6 mice compared to control, and this appears to be a T cell dependent mechanism, that is not perturbed by antibiotics (see comments below). TB meningitis samples from humans confirms an expanded CD4⁺ OX40 population exists, suggesting this population may have relevance to human TB.

In summary this manuscript contains some novel findings and promise but there remain some important issues to address.

1. The Nur77-GFP model can lead to rapid expression of GFP, but GFP will remain expressed for several days after the termination of TCR signals. As such it captures more the recent history (previous 3-4 days) rather than more 'real-time' TCR engagement which can be achieved by other tools within the field (Nur77-Tempo, Nr4a3-Tocky mice). The long half-life of GFP means that strongly self-reactive or tonic-signalling cells (e.g. Treg) can express significant levels of GFP even in peripheral tissues. The authors should acknowledge that this is a clear limitation of the study.
2. I appreciate that the authors split the data into Nur77 high and low populations, but this

will inherently bias towards cells receiving different TCR signal strengths (and/or frequency). Many of the gene sets described in Figure 2 are indicative of strong TCR signalling (OX40, GTR, IRF8, IL-21, see PMID: 34534438). I note that at the bulk RNAseq level Nr4a1 is only modestly differentially expressed suggesting Nur77-GFP levels are more reflecting historical TCR signals.

3. There is a strong Treg signature within the Nur77GFP hi population, which likely accounts for Treg activity within the lung. As such the transfer of Nur77-GFP hi cells will likely also carry across significantly more Treg cells which may alter the impact of the Nur77-GFP hi transfer group. What did % Foxp3+ look like in Nur77-GFP hi vs low vs bulk transfers? I strongly advise the authors to either cross Nur77-GFP hi cells with a Foxp3 reporter such as Foxp3-RFP, or to analyse the frequency of Foxp3+ T cells within their Nur77GFP hi and low groups. This is a critically important difference that may explain some of the biological mechanisms and should be experimentally addressed. In particular OX40 levels are very highly expressed on Foxp3+ T cells and as such Treg biology will be greatly modulated by agonistic OX40 antibody treatment.

4. The transferred cell populations for Figure 1E were sorted based on CD3 staining. This is likely to trigger/ affect the activity of this population following transfer as these clones are directly stimulatory. Were the bulk CD4+ T cells in Figure 1E also stained for CD3 before transfer? This is an important clarification that may alter the behaviour of the cells.

5. Given that the authors have tetramer staining, I think that showing GFP expression profiles of Nur77GFP and other markers in total ESAT6 and Ag85B should be included in the manuscript.

6. The conclusion about OX40 agonism and antibiotics is somewhat premature without performing longer experiments. These experiments should either be included, or the conclusions toned down. If Abx and OX40 agonism leads to better long term control this would be a significant finding and it is a slight shame that this has not been further explored.

7. Figure 1A – it would be informative to see a summary of Nur77-GFP hi cell frequency to assess the variation across mice. Also the flow cytometry plots in Fig 1A are unusually shifted into the negative on the biexponential. Typically, all T cells have some background Nur77-GFP expression in steady state due to tonic signal/ self-peptide MHC interactions, so please check the data/ compensation in Figure 1A.

Response to reviewers shown in BLUE:

Reviewer #1

“In this manuscript, Gress et al use Nur77-GFP reporter mice to characterize CD4 T cells that have recently seen Ag in the lungs of Mtb infected mice. They find that GFP-high cells are highly enriched in the in the tissue parenchyma compared to GFP-low cells. Moreover GFP-high CD4 T cells are more protective upon transfer into infected recipient mice. The authors use two different transcriptomic approaches, including a scRNAseq analysis, to characterize the unique properties of Nur77-GFP-high cells. The authors find that Nur77-GFP-high cells express high levels of the costimulatory molecule OX-40, so the authors explore the possibility of targeting OX-40 with agonist mAbs as a host directed therapy (HDT) strategy. It is found that cross-linking OX-40 in Mtb infected mice reduces bacterial loads and extends host survival. Lastly, the authors perform scRNAseq analysis of bulk T cells purified from the cerebrospinal fluid of individuals with HIV-TB co-infection associated meningitis and find the expression of OX-40 on CD4 T cells. The authors major conclusions are that CD4 T cells express OX-40 at sites of Mtb infection, and this costimulatory molecule may be a target for TB HDT. This manuscript contains a lot of transcriptomic data on T cells in tuberculosis, which are valuable data sets. There is interest in better understanding the nature of T cells that actively recognize Ag at sites of infection. Moreover, there is interest in identifying potential targets for HDT of TB. There are, however, concerns regarding the novelty of the major conclusions regarding OX-40 as well as concerns with the study design.”

Major concerns

1) The two major conclusions are that T cell express OX-40 and that it is a target for HDT of TB. The expression of OX-40 on CD4 T cells during Mtb infection, however, is not novel or unexpected.

We thank the reviewer for their constructive comments. Respectfully, we would counter that there is significant novelty in the finding of high levels of OX40 expression by CD4 and minimal expression by CD8 T cells recruited to the site of TB infection, in both mice and humans. Although this costimulatory receptor is known to be upregulated by effector T cells, especially after *ex vivo* antigen receptor stimulus, there is no prior study that investigates to what extent it is expressed *in vivo* at the site of TB infection, in mice or humans, and by which T cell population.

One of the most novel findings of the manuscript relates to the potential of treating tuberculosis by boosting CD4 T cell responses by co-stimulating OX-40 with cross-linking antibodies, but the impacts of OX-40 agonism were minimal. The difference in bacterial loads after anti-OX-40 is 2-fold at best, the survival experiment in 5B seems to represent a single experiment with n=3 mice, and OX-40 cross linking didn't enhance bacterial control as an adjunctive to antibiotic therapy. This seems to be a relatively weak case for targeting OX-40.

We appreciate the reviewer's concern regarding the robustness of the protective effect of OX40 agonism in Mtb infection. To strengthen the case that OX40 may be an

important target for host-directed immunotherapy in TB, we have repeated these treatment experiments, improved by control groups receiving isotype matched antibody injections, including CFU experiments with 5 additional mice per group and survival experiments containing 4-5 mice per group. These data are shown in the revised Figures 5B-C, 6B, and 7B.

Integrating these additional experimental data strongly confirms our initial finding that OX40 agonist antibody treatment for 2 weeks from week 4 to 6 of infection reduces the bacterial burden in the lungs by ~50%, an effect size that we feel is impressive for a short course of host-directed therapy.

We have also performed replicates of our initial experiments investigating adjunctive OX40 agonism co-administered with the antibiotics isoniazid and rifampin. These experiments are also better controlled with isotype antibody treated groups for comparison. These data indicate that the short-term antibacterial effect of OX40 agonism for 2 weeks is similar to the treatment effect that we observed with 2 weeks of isoniazid and rifampin provided in the drinking water, data shown in revised Figure 7B.

The addition of these well-controlled experimental replicates, along with longer term experimental time points 1 month and more after antibody treatment (revised Figures 6 and 7), help illustrate that the combination of these two treatment strategies offer an additive benefit on bacterial load reduction, with the group of mice receiving OX40 agonism + antibiotics having the lowest bacterial burden of all groups during the month following antibody treatment.

Have the authors tried agonizing OX-40 in other more susceptible mouse models including SP140^{-/-} mice?

We also are intrigued by the hypothesis that host-directed immunotherapy targeting OX40 or other costimulatory proteins expressed by activated CD4 T cells at the site of TB infection may help compensate for host immune susceptibilities, such as observed in hyper-susceptible strains of mice. We look forward to pursuing this question in future studies.

2) To compare cells that are Ag-stimulated vs. not stimulated, the authors gated on GFP-high or GFP-low cells. There are several limitations to this approach that are could have been improved on. Firstly, GFP-high cells are enriched in the lung parenchyma, so the differences found in this manuscript largely reflected the qualities of intravascular vs parenchymal T cells. which are populations that the authors point out have already been extensively previously described in Mtb infected mice. A much more informative comparison would have been to compare GFP-high vs GFP-low cells among the iv stain negative cells. This way, T cells of similar differentiation states and tissue localization would be compared.

We have performed additional flow cytometry analyses of Nur77-GFP^{HI} and LO cells, limiting our analysis to the IV negative lung parenchymal compartment. We show in new

figure S1E that even when limiting this comparison to lung parenchymal CD4 T cells there is a significant difference between both the phenotype of GFP HI and LO cells, with a higher fraction of GFPLO cells expressing CX3CR1 and KLRG1; as well as antigen specificity, with a higher frequency of tetramer staining among GFPHI parenchymal cells than GFPLO parenchymal cells.

This data may be available in the scRNAseq exp shown in Fig 3, but it is not analyzed for this comparison.

We favor using flow cytometry to make this comparison because of the clear distinction of vascular vs parenchymal cells using the established protocol of IV injection of fluorophore-labeled CD45 antibody immediately prior to euthanizing mice. Because of the shallow per cell sequencing depth of CITE/scRNA-Seq, we can only clearly conclude that oligo+ cells are more likely to be localized to the vasculature. This technique has substantially less power to clearly define events from which we do not recover sequence of the injected oligo-conjugated CD45 antibody as parenchymal T cells. This oligo-negative population may also include vascular cells bearing oligo that are below the limit of antibody detection by sequencing.

Secondly, the comparison of bulk GFP+ vs GFP- cells is difficult to interpret due to the likely difference in the abundance of Ag-specific T cells in these two populations. I think it is safe to expect that non-specific bystander T cells will be greatly enriched in the reporter negative cells. In one of the studies the authors perform tetramer pull down enrichment prior to sequencing, but the percentage of cells that are Ag-specific after this approach is still expected to be relatively low. Therefore, the pull-down experiment does not sufficiently address the caveat. Thus, the most informative approach to this experiment would have been to first isolate Ag-specific T cells that are in the parenchyma and then compare reporter high vs low. For example, this could be done by FACS purifying tetramer+ cells or by crossing the reporter to a ESAT-6 specific TCR Tg mouse.

We agree with the reviewer's prediction that any CD4 T cells in the lungs that are not specific for Mtb antigens would be greatly enriched among the Nur77-GFPLO compartment. To limit the number of these cells in our analyses, we have routinely performed gating of CD4 T cell populations on CD44hi effector cells, which expand significantly in the lungs after Mtb infection and are anticipated to be predominantly made up of cells specific for a variety of Mtb antigens in the context of the active infection at that site.

While the reviewer's suggested experiments may offer the potential benefit of more clearly identifying cells studied as Mtb-specific, they also have limitations that our approach can overcome:

First, limiting studies to T cells specific for a single tetramer or transgenic TCR would severely reduce the complexity and heterogeneity of the T cell population studied, raising new questions about how closely it reflects the complex, polyclonal endogenous

T cell response, a key goal of this study. Second, compared to the global CD44^{hi} CD4 T cell lung population, the smaller number of CD4 T cells staining with tetramers could also substantially raise the technical burden of the live sort and adoptive T cell transfer experiments we have successfully performed here. And finally, by focusing studies on just one or a few antigen specificities, we would not be able to discover TCRs potentially recognizing new Mtb epitopes that are associated with high levels of CD4 T cell activation and protective function, another goal of our ongoing work.

For these reasons we feel that the approach we have used in the current study offers a good balance of scientific rigor, technical feasibility, and opportunity to generate new knowledge.

3) The Nur77 reporter mice provide an obvious opportunity to examine the sites of T cells stimulation in the lungs, but this is not addressed. For example, do Nur77 reporter positive cells localize near bacteria infected cells?

We concur that this is a critically important question and ideal potential use for the Nur77-GFP reporter mice, which we intend to explore in future studies. Our data on the differential vascular vs parenchymal localization of GFP^{hi} vs LO cells offer a hint that CD4 T localization is likely a key determinant of activation status.

Minor points

1) In Figure 1B, it appears that Mtb-specific T cells express very little GFP. It is hard to tell from this data if ESAT-6 and Ag85b specific T cells are any different than other T cells in the lungs. A simple FACS plot of tetramer vs reporter would help the reader better evaluate the data. Just how robust is this reporter in identifying Ag-stimulated cells?

The data in Figure 1B illustrate the percentage of CD44^{hi} CD4⁺ T cells specific for ESAT-6 peptide or Ag85B peptide that are GFP^{hi} at day 28 post-infection. This percentage is similar to the percentage of GFP^{hi} cells among total CD44^{hi}, CD4⁺ T cells in the lungs. Although this percentage may vary slightly between individuals and from experiment to experiment as shown in new supplemental figures S1A and S1B, it is typically between 10-25%.

2) In Figure 1G, why do all the donor cells migrate into the parenchyma after transfer? It would be expected that some cells would be iv⁺ after transfer.

Although y-axis scaling in the original Figure 1G made this difficult to appreciate, we did observe both vascular and parenchymal localization of some donor cells, in similar proportions of both the GFP^{hi} and GFP^{lo} transfers. We have revised the original figure with logarithmic y-axis scaling to make this clearer.

3) I appreciate the attempt to confirm their murine model data in human T cells obtained from an infection site in patients. However, it is not clear how CSF cells from TB/HIV meningitis relate to lungs of Mtb infected mice.

We thank the reviewer for highlighting this potential caveat and have added text to the discussion to reflect this.

4) The data in 1H show a 10-fold reduction in bacterial loads just 1 weeks after transfer. This seems like a much larger effect than expected. This experiment would benefit from a control group did not receive T cells.

In a revised Figure 1H, we have included a group of mice that received no T cell transfer for comparison with the mice that received either Nur77-GFPHI or LO cells.

5) There is no analysis of GFP+ and GFP- CD8 T cells. Given the increasing interest in the field in the poor ability of CD8 T cells to recognize Mtb infected macrophages, it seems this is a missed opportunity.

Like the reviewer, we are highly interested in using the Nur77-GFP reporter to better define the CD8 T cell response against Mtb. However, for the sake of focus, we have chosen to restrict this manuscript to our studies of CD4 T cell biology. We have included new data in Figure S1D that compares the expression frequency and kinetics of Nur77-GFP by CD8 and CD4 T cells throughout infection.

6) The gating strategy for Nur77-GFP-high and low are not well defined, as it is not clear why the top 1/3 and bott 1/3 were chosen for downstream analysis when the gating in 1A shows two populations.

Because of the slight variation in Nur77-GFP expression from mouse to mouse and experiment to experiment, as well as the continuum of signal with a lack of a clearly defined break point in GFP signal to define recently activated and not recently activated cells, for sorting experiments and those involving direct comparison between GFPHI and LO, we used gates based on the highest and lowest 1/3 of GFP signal.

7) Figure 5 only shows bulk T cell responses and does not examine the frequency of antigen-specific T cells or functional changes after OX40 treatment.'

We have added additional data on the functional/phenotypic changes that occur among Mtb-tetramer specific and CD4 T cells after OX40 agonist treatment in the revised Figures 5 and S4, finding that these are recapitulated in the bulk polyclonal CD44hi CD4+ effector T cell population.

Reviewer #2 (Remarks to the Author):

In the manuscript titled "Activated CD4 T cells in tuberculosis express OX40, a novel target for host directed immunotherapy" Gress et al. utilize a Nur77-GFP reporter mouse to identify recently activated CD4+ T cells at the site of Mtb infection. This reporter mouse offers a tool to identify activated CD4+ T cells that are antigen specific for Mtb

without the limitations of needing to select for epitope specificities. The authors demonstrate that the Nur77-GFP^{hi} T cell subset at 4 weeks post-infection is protective against Mtb infection. They used several methods (bulk RNAseq, scRNAseq, scTCR sequencing) to perform a thorough characterization of this cell population which shows a heterogeneous mix of effector T cells with a striking upregulation of the costimulatory molecule OX40. They show how a OX40 agonist treatment can improve disease outcome and relate their findings to human data where OX40⁺ CD4⁺ T cells were identified in the cerebrospinal fluid of patients with HIV-associated TB meningitis. This manuscript is the first to describe OX40 as a marker for Mtb-specific T cells and to suggest OX40 agonism alone as a viable immunotherapy for treatment of tuberculosis.

Major concerns:

1. The reporter mouse used in this paper is an effective tool to study recently activated T cells following an infection. However, the authors did not show data to properly validate the model. Fig 1A should include an uninfected and infected wildtype control stained with a FACS Nur77 antibody to show that the transgene does not impact TCR activation at baseline, and also adequately represents the regular numbers of Nur77 T cell activation. Furthermore, the authors use the timepoint of 4 weeks post-infection for the entirety of the manuscript citing this as the timepoint at which T cells begin to control Mtb growth from Mogue et al (2001). However, Mogue et al. report day 20 post-infection as the timepoint where T cell responses reduce bacterial growth. Despite this, the authors should definitively show that within their model system, day 28 is a timepoint at which protective T cell responses are observed, that way, the comprehensive characterization of T cell responses on this day will be proven relevant. Thus, lung CFUs should be shown as a kinetic following infection. In addition to this, due to the transient nature of Nur77, the authors should show Nur77 expression overtime following infection to perhaps indicate a peak T cell activation at the chosen timepoint, in parallel with a reduction or control of bacterial growth.

We thank the reviewer for suggesting these validation experiments and have added the requested data showing the uninfected and infected wild type control mice (Fig 1A), the percentage of CD4 and CD8 T cells expressing GFP over time, which peaks around day 28 (Figure S1D), and the progression of lung CFUs as a kinetic over the course of infection of Nur77GFP mice (Figure S1C). These data support the choice of day 28 as a timepoint at which T cell activation is at a peak and protective T cell responses are observed in the Nur77-GFP mouse.

2. In this study, the authors use a Nur77-GFP reporter mouse to characterize activated vs bystander CD4⁺ T cell following Mtb infection. The Nur77-GFP^{hi} cell subset is described as Mtb-specific activated T cells and shown to have significantly increased effector function and protective capacity. The Nur77-GFP^{lo} and -GFP^{hi} are gated on CD3⁺CD44⁺ T cells, respectively representing activated and bystander subsets. The terminology used to describe the Nur77-GFP^{lo} and -GFP^{hi} cells is misleading since, by definition, CD44⁺ T cells are considered “activated”. The Nur77-GFP^{hi} T cells should rather be described as “recently activated”, considering the transient expression of Nur77 following TCR stimulation.

We appreciate this distinction and have adjusted the terminology used to refer to the GFPHI and LO subsets, removing use of “bystander” throughout the manuscript as suggested by the reviewer.

Furthermore, based on the presented gating strategy, CD44+ Nur77-GFPlo would include CD4+ memory T cells (effector, central and resident) that are present in the lungs and not antigen-specific for Mtb (Baaten et al. 2010). Therefore, the results from the comparison of Nur77-GFP^{hi} and Nur77-GFP^{lo} cells are expected. It is predictable that recently activated antigen-specific effector T cells (Nur77-GFP^{hi}) are more protective against Mtb than a heterogenous subset of memory/effector and antigen-specific/non-specific T cells. Although the experiments shown in Fig 1D-1H are important to validate this aspect of the model, the authors should highlight the underlying bias that grants Nur77-GFP^{hi} a more protective phenotype rather than presenting the described subset as a distinct and protective cell population.

We concur that an unknown proportion of the GFP^{lo} cells could be made up by cells that are not specific for Mtb antigens. This is a potential limitation, that prevents an overly conclusive characterization of the GFP^{lo} subset. Because of this important caveat, we have only intended to use the GFP^{lo} subset as an intra-mouse comparator population to enable characterization of the far more conclusively defined subset of GFP^{hi} cells, which we and others have demonstrated, represent those cells receiving high level antigen receptor stimulation within the prior 6-48 hours (Moran et al, 2011).

3. The study shows that a OX40 agonist treatment helps drive CD4 effector T cell responses thereby conferring protection against Mtb protection. Hence, the authors suggest further investigation into the clinical applications of OX40 agonism for treatment of tuberculosis. However, the data fails to show an additive effect of the combined treatment of antibiotics and OX40 agonism which represents the most relevant condition clinically. We suggest repeating this experiment with longer timepoints to ascertain potential long-term bacterial reduction or control. The reviewer also suggests comparing OX40 agonism to PD-1 blockade therapy to demonstrate the benefit of OX40 agonism compared to a widely used immunotherapy. In conjunction, seeing as PD-1 blockade results in various adverse effects (ie. IFN γ dysfunction), the authors could show that OX40 agonism does not have such outcomes and should therefore be considered as a more viable immunotherapy.

As suggested by the reviewer, we have completed additional experiments to further explore the protective benefit of OX40 agonism, and have compared it directly to PD-1 blockade. With increased group sizes and better control groups all receiving isotype control antibody injections, we have found that the short-term effect of OX40 agonism is robust, reducing CFUs after 2 weeks of treatment. This effect is improved by the addition of conjunctive antibiotic therapy with isoniazid and rifampin over the same time period- when compared to mice receiving isotype control injection and antibiotics as shown in revised Figure 7.

In contrast to PD-1 blockade which caused early post-treatment mortality in several mice (revised Figure S4B), we saw no apparent detrimental effects of treatment including no weight loss during or after treatment (revised Figure S4C). PD-1 blockade also had no significant impact on overall survival and did not increase CD4 or CD8 T cell counts, or reduce CFUs in the post-treatment window, as we observed for OX40 agonism (revised Figure 5).

Minor concerns:

Methods:

- The standard dose of aerosolized Mtb is 50-100 CFUs. A dose of 100-200 CFUs is used in this study and should therefore not be called a “low dose”, but a standard dose or high dose.

We appreciate that this is a meaningful distinction and have modified the text to more accurately reflect the standard infection dose used in this study.

Figure 1:

- Fig 1D-1H: There is inconsistency in the experimental setup and readouts of the two adoptive transfers.

o What is the reasoning behind transferring the T cells in the TCR α ^{-/-} recipient mice 1 week post-infection when 4 weeks is the timepoint of transfer into the wildtype recipient and the timepoint at which T cell responses are claimed to be protective during infection.

o Ideally, survival data and CFU data should be shown for both adoptive transfers.

o A no transfer control should be included in the adoptive transfer in wildtype recipients.

o The presence of Nur77-GFP cells in the vascular and parenchymal compartment of the lungs should be shown for the adoptive transfer in TCR α ^{-/-} recipients. The lymphopenic environment could be impacting the proliferation and localization of transferred T cells.

The different experimental setups in these two experiments reflect the different questions being asked in each context.

In the case of TCR α ^{-/-} recipients, we have used a previously established and published model for comparing the functional protection provided by different effector CD4 T cell subsets live sorted from the lungs of Mtb infected mice. In this context, as few as 4×10^4 cells from the lungs of infected mice have been previously demonstrated to be able to provide protection to highly susceptible TCR α ^{-/-} recipients when adoptively transferred 1 week into the infection (Sakai et al, 2016). The readout used to measure this protection is the difference between the survival of the two groups. We did not quantitate the difference in lung CFUs between the donor groups in this context partly because we do not hypothesize that the adoptively transferred population will be able to compensate for elevated bacterial loads due to the complete deficiency of α/β T cells in these hosts.

For wild type recipients, we sought to determine whether the difference in protection provided could be due to a difference between the trafficking of the GFPHI and LO subsets into lung parenchyma. We used a 1-week time point after adoptive transfer to assess the ability of donor cells to incorporate into an ongoing native infection context. The findings that GFPHI cells still provide protection, in this case measured as CFUs 1 week after adoptive transfer, despite no significant difference in the number of cells localizing to the parenchyma indicates that the difference in protection is likely provided by other characteristics differing between the GFPHI and LO subsets. This effect could be partially attributable to impacts of donor CD4 T cells on the endogenous CD4 or CD8 T cells in the wild type host.

- Fig 1G: There appears to be no detectable levels of Nur77-GFPlo or Nur77-GFPHI cells in the vasculature of the recipient mice which seems unusual if the T cells were transferred intravenously. Additionally, Nur77-GFPlo cells are described to usually be found in the vascular compartment, which suggests that this adoptive transfer is not adequately recapitulating the true localization of the T cell subsets. Therefore, the conclusions made about the protective capacity of Nur77-GFPlo cells could be incorrect.

As in our response to Reviewer 1, we did observe both vascular and parenchymal localization of some donor cells, in similar proportions of both the GFPHI and GFPLO transfers. We have revised the original figure to make this clearer.

Figure 4:

- In the paper that first described this reporter mouse, Moran et al. show that Nur77-GFP expression is also a sensitive reporter for the measure TCR signal strength. Describing the three major clonotypes as the ones who can confer preferential activation could be erroneous since Nur77-GFPlo T cells also show antigen-specificity and were therefore at some point also activated. The authors should describe these findings as the clonotypes that induce strong TCR stimulation, rather than activation.

We agree with the reviewer that Nur77GFPHI cells also likely reflect those receiving higher signal strength via the TCR due to higher antigen affinity, and we have modified the text throughout to incorporate this possibility.

- This data shows that the clonotypes between Nur77-GFPlo and -GFPHI are similar which indicates that a large portion of Nur77-GFPlo T cells are Mtb-specific. In light of this, the term “bystander” used to define Nur77-GFPlo cells should be revised as bystander T cells are typically defined by activation of inflammatory signals without antigen recognition.

We have removed all references to “bystander” cells; instead using the terms “recently activated” and “not recently activated” cells to refer to GFPHI and LO subsets, respectively.

Figure 5:

- Fig 5D, 5E: An isotype control should be included just as in 5C.

We have repeated all OX40 agonist experiments using control group mice treated with Isotype control antibody at the same doses, infection time points, and frequencies.

- The axes on both CFU graphs should be the same.

We have standardized the logarithmic y-axis scale for key CFU plots to enhance comparability throughout the manuscript.

- Fig 5D: The sentence in the figure legend “Cell numbers assessed using flow cytometry with counting beads, gated on live CD3+CD44+ CD4+ or CD8+ cells” is misplaced and should follow the 5D description not 5E.

We have corrected this.

- Fig 5E: The authors specify that that the antibiotic treatment and OX40 agonist treatment both lasted two weeks. However, the pretreatment group and treatment group are 11 days apart, indicating that the treatment regiment could not have lasted two weeks.

In our antibody treatment experiments, we have given 4 doses of antibody twice weekly for two weeks, with antibody treatments on days 28, 31, 35, and 38 post-infection. The final CFU harvest is performed after the final dose, typically on day 39-42 post-infection. We have clarified the text to reflect this protocol.

Reviewer #3 (Remarks to the Author):

In this manuscript by Gress and colleagues they investigate T cell activation within the lung parenchyma in response to murine TB infection. They employ the established Nur77-GFP model (see comments below), which reports activity of Nr4a1 (an immediate early TCR-activated gene used as a proxy for T cell activation). Infection of mice leads to a large increase in Nur77-GFP^{hi} T cells, i.e. recently strongly TCR-stimulated cells, that they further interrogate in comparison to Nur77-GFP^{lo} counterparts. Nur77-GFP^{hi} T cells are more greatly enriched for antigen-specific T cells and have superior protective capacity compared to Nur77-GFP^{low} T cells. The authors cite this as evidence for a functional difference in the population (see comment below). Nur77GFP^{hi} cells comprise a heterogeneous population of Treg and effector cells, and Nur77GFP^{hi} have many hallmarks of strong TCR signalling. Nur77GFP^{hi} cells have higher levels of co-receptors (such as OX40, GITR, 41BB) and co-inhibitory receptors (Lag3, CTLA-4). Whilst Nur77GFP^{hi} and low cells share similar dominant clones, the Nur77GFP^{hi} fraction shows greater expansion of the top three dominant clonotypes. The authors

show that OX40 is a selective marker of Nur77-GFP hi cells, that is largely restricted to CD4 but not CD8 T cells. OX40 agonism enhances survival of TB infected B6 mice compared to control, and this appears to be a T cell dependent mechanism, that is not perturbed by antibiotics (see comments below). TB meningitis samples from humans confirms an expanded CD4+ OX40 population exists, suggesting this population may have relevance to human TB.

In summary this manuscript contains some novel findings and promise but there remain some important issues to address.

1. The Nur77-GFP model can lead to rapid expression of GFP, but GFP will remain expressed for several days after the termination of TCR signals. As such it captures more the recent history (previous 3-4 days) rather than more 'real-time' TCR engagement which can be achieved by other tools within the field (Nur77-Tempo, Nr4a3-Tocky mice). The long half-life of GFP means that strongly self-reactive or tonic-signalling cells (e.g. Treg) can express significant levels of GFP even in peripheral tissues. The authors should acknowledge that this is a clear limitation of the study.

We have added text revisions to discuss the implications of the time dependent expression of the GFP reporter and the potential advantages and limitations of this particular mouse model vs other available lymphocyte activation reporters.

2. I appreciate that the authors split the data into Nur77 high and low populations, but this will inherently bias towards cells receiving different TCR signal strengths (and/or frequency). Many of the gene sets described in Figure 2 are indicative of strong TCR signalling (OX40, GITR, IRF8, IL-21, see PMID: 34534438). I note that at the bulk RNAseq level Nr4a1 is only modestly differentially expressed suggesting Nur77-GFP levels are more reflecting historical TCR signals.

As in response to Reviewer 2, we agree that when compared to Nur77-GFPLO cells, Nur77-GFPHI cells will include both cells that are more recently activated as well as potentially those receiving a stronger TCR signal strength, and we have revised our discussion of these results to be inclusive of this possibility.

3. There is a strong Treg signature within the Nur77GFP hi population, which likely accounts for Treg activity within the lung. As such the transfer of Nur77-GFP hi cells will likely also carry across significantly more Treg cells which may alter the impact of the Nur77-GFP hi transfer group. What did % Foxp3+ look like in Nur77-GFP hi vs low vs bulk transfers? I strongly advise the authors to either cross Nur77-GFP hi cells with a Foxp3 reporter such as Foxp3-RFP, or to analyse the frequency of Foxp3+ T cells within their Nur77GFP hi and low groups. This is a critically important difference that may explain some of the biological mechanisms and should be experimentally addressed. In particular OX40 levels are very highly expressed on Foxp3+ T cells and as such Treg biology will be greatly modulated by agonistic OX40 antibody treatment.

Our single cell RNA Sequencing analysis as reported in Figure 3 indicates that Tregs are clearly enriched in the Nur77-GFP^{HI} population, though they make up only a small fraction of the total population, represented by a cluster of Foxp3-enriched cells. We have performed flow cytometry analyses with intracellular FOXP3 staining that confirms a small but statistically enriched population of Tregs, accounting for approximately 5% of the Nur77-GFP^{HI} population (revised Fig S5)

4. The transferred cell populations for Figure 1E were sorted based on CD3 staining. This is likely to trigger/ affect the activity of this population following transfer as these clones are directly stimulatory. Were the bulk CD4⁺ T cells in Figure 1E also stained for CD3 before transfer? This is an important clarification that may alter the behaviour of the cells.

The bulk CD4 T cells used as comparisons for the adoptive transfer experiments were not stained with anti-CD3. We have modified the methods to clearly state this.

5. Given that the authors have tetramer staining, I think that showing GFP expression profiles of Nur77GFP and other markers in total ESAT6 and Ag85B should be included in the manuscript.

Figure 1B shows the fraction of ESAT-6 and Ag85B specific, CD44^{HI}, CD4⁺ effector T cells that are Nur77-GFP positive at day 28 post-infection. Revised Figure S1E indicates the fraction of IV- parenchymal Nur77-GFP^{HI} and LO cells that are tetramer⁺.

6. The conclusion about OX40 agonism and antibiotics is somewhat premature without performing longer experiments. These experiments should either be included, or the conclusions toned down. If Abx and OX40 agonism leads to better long term control this would be a significant finding and it is a slight shame that this has not been further explored.

We have performed additional experiments including longer term time points post antibody treatment with and without antibiotic therapy. These are included in revised Figures 6 and 7. These studies have revealed an additive effect of antibiotics and short term OX40 agonist immunotherapy, with mice receiving both antibiotics and OX40 agonist having the lowest CFUs after 2 weeks of treatment. Even 1 month after ceasing OX40 agonism the effect of reduced CFUs in this dual treated group is observed compared to isotype control/antibiotic-treated mice. This is likely the result of persistent changes in CD4 T cell population size and a lasting shift away from the terminal differentiation phenotype (CX3CR1/KLRG1-expression) observed at this time point.

7. Figure 1A – it would be informative to see a summary of Nur77-GFP^{hi} cell frequency to assess the variation across mice. Also the flow cytometry plots in Fig 1A are unusually shifted into the negative on the biexponential. Typically, all T cells have some background Nur77-GFP expression in steady state due to tonic signal/ self-peptide MHC interactions, so please check the data/ compensation in Figure 1A.

We have presented the variation in Nur77-GFP expression among mice within each experiment, as well as across 8 separate experiments in new Figure S1B.

REVIEWER COMMENTS

Reviewer #1 (Remarks to the Author):

The authors have addressed my most critical concerns.

Reviewer #2 (Remarks to the Author):

The authors experimentally addressed all my concerns and I have no further comment.

Reviewer #3 (Remarks to the Author):

The authors have responded to most of my major criticisms adding in new data and clarifying some points.

I do have one remaining major criticism regarding the statistical analyses in figures 5-7. The inclusion of new data means that often more than two treatments are being tested (e.g. Isotype vs anti-PD1 vs anti-OX40) or two conditions over time (requiring a two-way ANOVA analysis). From my reading all these statistical tests are individual t tests, and from I see no evidence of any correction for multiple testing. I would have thought that the most appropriate test in Figure 5-E would be a One-way ANOVA with post-hoc testing. Similarly, the time course data in Figures 6 and 7 should be analysed by two way anova factoring in the variables "Treatment" and "time", with post hoc testing of treatment if this is a significant source of variation in the model.

If the above can be satisfactorily addressed, then I believe the manuscript is suitable for publication.

Reviewer #4 (Remarks to the Author):

In this manuscript, Gress et al. demonstrated recently activated antigen-specific T cells expressing OX40 showed protective immunity at sites of infection during early TB progression by employing a murine Nur77-GFP model system. The authors concluded

Nur77-GFPHI CD4 T cells are more protective than Nur77-GFPLO cells against Mtb infection. Ultimately, the authors identified OX40 agonism as a therapeutic potential in murine TB models. In addition, the authors found CD4 T cells, not CD8 T cells, expressing OX40 existed in CSF of patients with HIV-TB meningitis, which indicating clinical relevance. Although the authors identified a promising target of HDT against TB in particular the early and primary of Mtb infection, there are remaining concerns regarding the used models, methodologies, and central conclusions.

Major concerns:

1. The functional implications of Nur77 and OX40 expression levels on T cells could be modulated by the specific T cell subtypes. While the authors have explored the initial impact of OX40 agonism with a particular rationale, the translation of OX40 agonism into host-directed therapy (HDT) within a clinical context warrants further consideration. The potential application of OX40 agonism as an HDT strategy may necessitate the identification of suitable biomarkers to gauge therapeutic efficacy. It is noteworthy that prolonged presence of antigen-specific CD4 T cells, even though protective in the short term, could potentially exert detrimental effects on the host by inducing tissue damage during a chronic state. Please discuss this point.

2. In Figure 8, the reviewer acknowledges the discernment of OX40 expression primarily within the CD4 T cell subset rather than CD8 T cells. Nevertheless, it is imperative to consider the gravity of HIV-associated TB meningitis, which represents the most severe form of TB. This context may underscore differential roles of OX40+CD4 T cells between murine TB models and human severe TB form. As a consequence, the reviewer expresses a contemplation regarding the clinical significance underpinning this observation.

The fundamental inquiry arises with regard to the substantive importance attributed to the discovered correlation between OX40+ CD4 T cells and TB protection. An intriguing speculation is posited: the hypothetical existence of this protective T cell population may be compromised in instances of severe TB pathology. In lieu of the prominently anticipated OX40+ CD4 T cell cohort, alternative immunological elements, such as the type I interferon (IFN) signature and FoxP3+ regulatory T cells (Treg), may exert a more conspicuous association with OX40 expression. A comprehensive deliberation of this aspect is warranted,

fostering an intricate discourse on the plausible intricacies of OX40-mediated immunity in the context of TB pathogenesis.

3. I appreciate the clear separation of Nur77-GFP high and low cells for their high purity. As demonstrated in Figure 4 through T cell trajectory analysis, whether Nur77 is Mtb antigen-specific or if Nur77HI and Nur77LO exhibit distinct clonotypes, there appears to be a significant overlap in TCR clonotypes between the Nur77-GFPLO and Nur77-GFPHI populations. Therefore, it is possible that the transition from GFP expression at low levels to high levels may occur transiently (at a specific time point, such as 4 weeks post-infection). An explanation for this phenomenon is warranted. Although the demonstrated the recent activation of T cells in an antigen-specific manner through Ag85B peptide injection, it is important to note that in actual in vivo conditions, the expression of GFP may vary due to factors such as the cytokine milieu. Given the potential differences in transient expression levels of GFP, the results of restimulation could provide insights into the functionality. This would help in understanding if the observed changes in Nur77-GFP expression from LO to HI have functional implications.

4. I appreciate the inclusion of extensive long-term survival experiments with regards to the effects of OX40 agonism and T cell transfer. Nonetheless, it is noteworthy to highlight the importance of conducting more detailed assessments in relation to these protection experiments. Specifically, there is a need for more precise evaluations, as indicated by the histopathology scores presented in Figures 1, 5, 6, and 7. This additional level of analysis would contribute to a more comprehensive understanding of the outcomes observed in the study.

5. It is crucial to address the noteworthy observation of the rapid elevation of T cell exhaustion markers such as KLRG1 in the absence of antigen experience in GFPLO T cells at early time point of Mtb infection. This phenomenon raises considerable curiosity and therefore please provide a discussion.

Minor comments;

1. In Figure 1, the authors assert that Nur77-GFPHI CD4 T cells exhibit enhanced functional protection. To substantiate this claim, it is recommended to characterize the antigen-specific protective cytokine profile within the sorted Nur77-GFP CD4 T cell populations at

protein levels if possible.

2. In Figure 2, an unexpected observation arises where the expression of Nr4a1, along with EGFP, is markedly limited in comparison to the OX40-expressing cell population, as evidenced in the RNA sequencing analysis. Additionally, the non-overlapping nature of these populations is evident in Figure 2D. It is imperative to provide a clear rationale for this distinct pattern, which appears to delineate these entities as separate populations, despite the co-expression of the two markers. Furthermore, within the Nur77-GFPLO CD4 T cell subset, it is noteworthy that even parenchymal Tconv effect cells exhibit noteworthy OX40 expression, as illustrated in Figure 3B. This finding should be elaborated upon to elucidate the potential implications of such high OX40 expression within this specific subpopulation of Nur77-GFPLO CD4 T cells.

3. In Figure 5, it is evident that OX40 agonistic mAb treatment imparts a sustained long-term survival benefit. However, it is noteworthy that the impact of OX40 agonistic mAb treatment on the reduction of CFUs appears to diminish at the 42-day post-treatment time point, as depicted in Figure 6. This diminishing trend over time contrasts with the enduring presence of an effector CD4 T cell population in the parenchyma, as demonstrated in Figure 6 and 7. This phenomenon warrants elucidation to provide a comprehensive understanding of the observed results. Moreover, it is crucial to acknowledge that a precise assessment of the long-term effects of OX40 mAb treatment necessitates histopathology analysis. This comprehensive evaluation will ascertain the sustained protective CD4 T cells by treating with OX40 mAb is actually protective.

4. Very minor comments on abstract: Please insert comma in line 28 (Page 2, After Mycobacterium tuberculosis (Mtb) infection,). Please delete comma line 35 (Page 2, A short).

Notes from reviewer about scRNA sequencing

Thank you for asking me one more time. Technically, there was no issue on the single cell RNA sequencing.

Just two minor comments:

1. In Figure 2 (Former minor comment #2), an unexpected observation arises where the expression of Nr4a1, along with EGFP, is markedly limited in comparison to the OX40-expressing cell population, as evidenced in the RNA sequencing analysis.
2. In Figure 4 (Former major comment #3), . As demonstrated in Figure 4 through T cell trajectory analysis, whether Nur77 is Mtb antigen-specific or if Nur77HI and Nur77LO exhibit distinct clonotypes, there appears to be a significant overlap in TCR clonotypes between the Nur77-GFPLO and Nur77-GFPHI populations

I have just another minor comments on scRNA sequencing in CSF of HIV-TB patients.

It would be helpful if the authors indicate the number of the sorted live CD4+/CD8+ T cells individual patients in M&M section.

Response to reviewers and new manuscript revisions are shown in **RED**:

Reviewer #3 (Remarks to the Author):

The authors have responded to most of my major criticisms adding in new data and clarifying some points.

I do have one remaining major criticism regarding the statistical analyses in figures 5-7. The inclusion of new data means that often more than two treatments are being tested (e.g. Isotype vs anti-PD1 vs anti-OX40) or two conditions over time (requiring a two-way ANOVA analysis). From my reading all these statistical tests are individual t tests, and from I see no evidence of any correction for multiple testing. I would have thought that the most appropriate test in Figure 5-E would be a One-way ANOVA with post-hoc testing.

We thank the reviewer for recommending this statistical re-analysis to improve the rigor of this study. We have performed the recommended testing and include revised figures, figure legends and statistical methods sections indicating the updated statistical tests used. Importantly, these new analyses add further support to the original interpretation of the data presented.

For Figure 5C-E, we used one-way ANOVA to compare the results of the three treatments: isotype control, PD-1 blockade, and OX40 agonism, with an overall p value for comparing the means of the three groups reported in the figure 5 legend. P values shown in the figures represent post-hoc testing adjusted for multiple comparisons.

Similarly, the time course data in Figures 6 and 7 should be analysed by two way anova factoring in the variables "Treatment" and "time", with post hoc testing of treatment if this is a significant source of variation in the model.

If the above can be satisfactorily addressed, then I believe the manuscript is suitable for publication.

For Figure 7B and E, we used two-way ANOVA to assess whether an interaction exists between the two treatments (antibiotics and antibody) and determine whether this interaction is a significant source of variation in the model. We have revised these analyses in the figure, figure legend, and corresponding results section. For 7B CFUs, two-way ANOVA indicated that each treatment alone, as well as the interaction between the two, contribute statistically significant variation to the observed mean CFUs for each group, supporting our central conclusion that OX40 agonism offers additive benefit to antibiotic treatment alone. ANOVA p values and percent variance attributable to each are provided in the Figure 5 legend. P values shown in the figure represent those adjusted for multiple comparisons of isotype vs OX40 agonist antibody either with or without antibiotics.

For 7E comparison of %CX3CR1/KLRG1+ cells, we observed a significant impact of OX40 agonism compared to isotype control either with or without antibiotics, with p values adjusted for multiple comparisons. The two-way ANOVA in this case indicated a significant impact on variation due to both antibody and antibiotic individually, though the p value for the interaction was not significant. We have revised the figure, figure legend, and results section to reflect this.

For data in Figures 6 and 7 comparing isotype control or OX40 agonist, from either two or three time points, we suggest that the most appropriate statistical test are unpaired t-tests with

adjustment for multiple comparisons, specifically limiting statistical comparison to measurements between two treatment groups at the same time point. We have revised the figures and figure legends to reflect this more robust analysis, which also supports the original conclusions based on these data.

For these particular comparisons, we feel this statistical approach is more appropriate than two factor ANOVA because 1) we are focused on assessing differences within each time point, not between them, and 2) an assumption of ANOVA is that variables are independent between groups and within a group. Because a later time point is inherently dependent on events occurring at an earlier time point, applying time as a variable in standard ANOVA violates this assumption. Although repeated measures ANOVA can assess the interaction between time and another independent variable, this requires repeated measurements from the same individual, enabling paired comparisons of dependent variables over time. Since each group of mice was euthanized for each time point's measurements, this was not possible in our study.

Reviewer #4 (Remarks to the Author):

In this manuscript, Gress et al. demonstrated recently activated antigen-specific T cells expressing OX40 showed protective immunity at sites of infection during early TB progression by employing a murine Nur77-GFP model system. The authors concluded Nur77-GFPHI CD4 T cells are more protective than Nur77-GFPLO cells against Mtb infection. Ultimately, the authors identified OX40 agonism as a therapeutic potential in murine TB models. In addition, the authors found CD4 T cells, not CD8 T cells, expressing OX40 existed in CSF of patients with HIV-TB meningitis, which indicating clinical relevance. Although the authors identified a promising target of HDT against TB in particular the early and primary of Mtb infection, there are remaining concerns regarding the used models, methodologies, and central conclusions.

Major concerns:

1. The functional implications of Nur77 and OX40 expression levels on T cells could be modulated by the specific T cell subtypes. While the authors have explored the initial impact of OX40 agonism with a particular rationale, the translation of OX40 agonism into host-directed therapy (HDT) within a clinical context warrants further consideration. The potential application of OX40 agonism as an HDT strategy may necessitate the identification of suitable biomarkers to gauge therapeutic efficacy. It is noteworthy that prolonged presence of antigen-specific CD4 T cells, even though protective in the short term, could potentially exert detrimental effects on the host by inducing tissue damage during a chronic state. Please discuss this point.

We thank the reviewer for calling attention to this implication of our data and we have added text to the discussion as requested. As the reviewer notes in a comment below, our observation of long-term increased survival of mice treated with OX40 agonist, extending to months after treatment, suggests a net positive effect of this treatment as tested in this study.

2. In Figure 8, the reviewer acknowledges the discernment of OX40 expression primarily within the CD4 T cell subset rather than CD8 T cells. Nevertheless, it is imperative to consider the gravity of HIV-associated TB meningitis, which represents the most severe form of TB. This context may underscore differential roles of OX40+CD4 T cells between murine TB models and human severe TB form. As a consequence, the reviewer expresses a contemplation regarding the clinical significance underpinning this observation.

The fundamental inquiry arises with regard to the substantive importance attributed to the discovered correlation between OX40+ CD4 T cells and TB protection. An intriguing speculation is posited: the hypothetical existence of this protective T cell population may be compromised in

instances of severe TB pathology. In lieu of the prominently anticipated OX40+ CD4 T cell cohort, alternative immunological elements, such as the type I interferon (IFN) signature and FoxP3+ regulatory T cells (Treg), may exert a more conspicuous association with OX40 expression. A comprehensive deliberation of this aspect is warranted, fostering an intricate discourse on the plausible intricacies of OX40-mediated immunity in the context of TB pathogenesis.

We heartily agree with the reviewer and find their hypothesis intriguing. We have included new discussion regarding the detection of type I IFN signature among TNFRSF4 expressing CD4 T cells in the CSF. We have amended the discussion to include the finding of FOXP3 expressing clusters (also observed in the mouse among the Nur77-GFP^{HI} subset) to clarify and call greater attention to the potential similarities and differences between CD4 T cells in the CSF of human subjects with HIV-associated TB meningitis and the lungs of mice infected with Mtb.

3. I appreciate the clear separation of Nur77-GFP high and low cells for their high purity. As demonstrated in Figure 4 through T cell trajectory analysis, whether Nur77 is Mtb antigen-specific or if Nur77^{HI} and Nur77^{LO} exhibit distinct clonotypes, there appears to be a significant overlap in TCR clonotypes between the Nur77-GFP^{LO} and Nur77-GFP^{HI} populations. Therefore, it is possible that the transition from GFP expression at low levels to high levels may occur transiently (at a specific time point, such as 4 weeks post-infection). An explanation for this phenomenon is warranted. Although the demonstrated the recent activation of T cells in an antigen-specific manner through Ag85B peptide injection, it is important to note that in actual in vivo conditions, the expression of GFP may vary due to factors such as the cytokine milieu. Given the potential differences in transient expression levels of GFP, the results of restimulation could provide insights into the functionality. This would help in understanding if the observed changes in Nur77-GFP expression from LO to HI have functional implications.

We agree with the reviewer about the overlap between TCR clonotypes in Nur77-GFP^{HI} and LO cells from the same mouse and called attention to this in our original manuscript submission. We concur that this is an expected finding likely reflecting both the continuous spectrum of GFP expression we observed rather than two discrete populations, and the transient nature of Nur77-GFP expression- within 6-48 hours of antigen receptor stimulation, as first shown by Moran et al, 2011. We have added additional text to clarify this explanation in the results section.

We also agree that restimulation can sometimes clarify the functionality of effector T cells; however, for this study, we have specifically avoided using ex vivo restimulation as a technical strategy throughout. This is because it is contrary to our original goal to identify and characterize the subset of cells receiving antigen receptor at the site of Mtb infection in vivo.

4. I appreciate the inclusion of extensive long-term survival experiments with regards to the effects of OX40 agonism and T cell transfer. Nonetheless, it is noteworthy to highlight the importance of conducting more detailed assessments in relation to these protection experiments. Specifically, there is a need for more precise evaluations, as indicated by the histopathology scores presented in Figures 1, 5, 6, and 7. This additional level of analysis would contribute to a more comprehensive understanding of the outcomes observed in the study.

We are also very interested in understanding the contribution of OX40 agonism to histopathology and agree that this will likely be an important component of understanding the beneficial effect of this treatment. Counter to the reviewer's comment, we have not included any histopathology scores in this manuscript. Instead, we have focused our assessment on the targeted impact of OX40 agonist immunotherapy on bacterial burden, CD4 T cell function and

phenotype. We look forward to determining how the effects we have observed will affect broader histopathologic findings in future studies.

5. It is crucial to address the noteworthy observation of the rapid elevation of T cell exhaustion markers such as KLRG1 in the absence of antigen experience in GFPLO T cells at early time point of Mtb infection. This phenomenon raises considerable curiosity and therefore please provide a discussion.

In contrast to other models of T cell immunity, CX3CR1 and KLRG1 co-expression in the context of chronic Mtb infection of mice has been used as a defining feature of cells with a Th1 “terminal differentiation” rather than “exhaustion” phenotype. We hypothesize that the enriched expression of these markers among Nur77-GFPLO cells reflects a history of antigen receptor stimulation- though not recent- followed by downregulation of the Nur77-GFP reporter.

Terminally differentiated effector CD4 T cells that express high levels of CX3CR1/KLRG1 in the mouse model of TB have been observed by us and others (e.g. Sallin et al 2017) to preferentially localize to the vasculature, spatially isolated from lung parenchymal lesions where antigen is presumably the most available. Our data presented here are consistent with those prior results, and we have added an additional discussion point to the revised manuscript to put these findings into a clearer context.

Minor comments:

1. In Figure 1, the authors assert that Nur77-GFPHI CD4 T cells exhibit enhanced functional protection. To substantiate this claim, it is recommended to characterize the antigen-specific protective cytokine profile within the sorted Nur77-GFP CD4 T cell populations at protein levels if possible.

This is a very interesting point that we are also interested in exploring. In this manuscript we have focused on the transcriptional heterogeneity of effector CD4 T cells activated in vivo by antigen stimulation in the lungs, characterizing their localization, antigen specificity, TCR repertoire, and expression of key phenotypic surface markers. We have also assessed their functional capacity using several adoptive transfer approaches. Importantly, in contrast to cytokine production, none of these approaches depends on ex vivo restimulation.

As discussed above, this is an important technical point for the design of this particular study. Restimulation can provide information about the functionality of effector T cells, using intracellular staining with flow cytometry, ELISA, or ELISpot as a readout. However, for this study, we have avoided using ex vivo restimulation throughout. This is because it is contrary to our goal to identify and characterize the subset of cells receiving antigen receptor at the site of Mtb infection in vivo. Although this limits our ability to provide protein level data on the expression of cytokines, we feel that the transcriptional data on cytokine gene expression we have provided reveals valuable information about the functional cytokine capacities of these two different cell subsets.

2. In Figure 2, an unexpected observation arises where the expression of Nr4a1, along with EGFP, is markedly limited in comparison to the OX40-expressing cell population, as evidenced in the RNA sequencing analysis. Additionally, the non-overlapping nature of these populations is evident in Figure 2D. It is imperative to provide a clear rationale for this distinct pattern, which appears to delineate these entities as separate populations, despite the co-expression of the two markers. Furthermore, within the Nur77-GFPLO CD4 T cell subset, it is noteworthy that even parenchymal Tconv effect cells exhibit noteworthy OX40 expression, as illustrated in

Figure 3B. This finding should be elaborated upon to elucidate the potential implications of such high OX40 expression within this specific subpopulation of Nur77-GFPLO CD4 T cells.

The imperfect overlapping expression pattern of Nr4a1 and Tnfrsf4 is most likely a manifestation of multiple technical and biological phenomena. As requested by the reviewer, we have added text revision to the Results and Discussion to elaborate on this finding, summarized as follows:

Some differences are expected in the temporal expression pattern and detection threshold between for Nr4a1 transcript using scRNA-Seq and fluorescent signal of Nur77-GFP fluorescent protein using flow cytometry. Nr4a1/EGFP transcript is very likely expressed for a shorter duration than the reporter protein itself, which can explain an incomplete overlap of cells Nur77-GFPHI by flow and Nr4a1/EGFP-expressing by RNA-Seq. We hypothesize that some cells in Figure 2D that are Nr4a1/EGFP transcript positive, but Tnfrsf4 transcript negative may reflect the most recently activated cells that either have yet to fully upregulate Nur77-GFP reporter protein or represent an early stage of effector function or differentiation prior to abundant *Tnfrsf4*/OX40 expression. Finally, as discussed in the manuscript, because Nur77-GFP expression is detected as a spectrum of fluorescence signal, reflecting asynchronous T cell activation in vivo, as well as antigenic stimuli of differing TCR signal strength, it was not possible to perfectly define the cells that are Nur77-GFP+ or -; rather we could only compare the highest and the lowest 1/3 of GFP expression. We feel that this approach offered valuable information about the enrichment of certain markers among the activated effector CD4 T cell population in the lungs. Importantly, we show in Figure 3E that, when assessing dual expression of the two proteins, 32% of Nur77-GFPHI cells express OX40, compared to 4% of Nur77-GFPLO cells by flow cytometry. Our main conclusion from these data is that recently activated CD4 T cells are enriched for cells expressing the costimulatory receptor OX40.

3. In Figure 5, it is evident that OX40 agonistic mAb treatment imparts a sustained long-term survival benefit. However, it is noteworthy that the impact of OX40 agonistic mAb treatment on the reduction of CFUs appears to diminish at the 42-day post-treatment time point, as depicted in Figure 6. This diminishing trend over time contrasts with the enduring presence of an effector CD4 T cell population in the parenchyma, as demonstrated in Figure 6 and 7. This phenomenon warrants elucidation to provide a comprehensive understanding of the observed results. Moreover, it is crucial to acknowledge that a precise assessment of the long-term effects of OX40 mAb treatment necessitates histopathology analysis. This comprehensive evaluation will ascertain the sustained protective CD4 T cells by treating with OX40 mAb is actually protective.

We agree with the reviewer that the discrepancy between persistent phenotypic changes among the CD4 T cell population despite recovery of the lung bacterial burden after cessation of OX40 agonist antibody treatment is worthy of additional study. We hypothesize that these phenotypic changes confer enhanced protection independent of bacterial burden, conferring enhanced long-term survival despite chronic infection. We have added text revisions to the discussion to explicitly acknowledge the critical importance of future studies to assess how this occurs, which should certainly include, as the reviewer suggests histopathology analyses, as well as more comprehensive analyses of CD4 T cell function at later time points after cessation of immunotherapy.

4. Very minor comments on abstract: Please insert comma in line 28 (Page 2, After *Mycobacterium tuberculosis* (Mtb) infection,). Please delete comma line 35 (Page 2, A short).

We thank the reviewer and have made these suggested edits.

Notes from reviewer about scRNA sequencing

Thank you for asking me one more time. Technically, there was no issue on the single cell RNA sequencing.

Just two minor comments:

1. In Figure 2 (Former minor comment #2), an unexpected observation arises where the expression of Nr4a1, along with EGFP, is markedly limited in comparison to the OX40-expressing cell population, as evidenced in the RNA sequencing analysis.

Please see above response to minor comment #2.

2. In Figure 4 (Former major comment #3), . As demonstrated in Figure 4 through T cell trajectory analysis, whether Nur77 is Mtb antigen-specific or if Nur77HI and Nur77LO exhibit distinct clonotypes, there appears to be a significant overlap in TCR clonotypes between the Nur77-GFPLO and Nur77-GFPHI populations

Please see above response to major comment #3.

I have just another minor comments on scRNA sequencing in CSF of HIV-TB patients.

It would be helpful if the authors indicate the number of the sorted live CD4+/CD8+ T cells individual patients in M&M section.

We sorted 5,000 cells from each population. This information is included in the original submission of the manuscript in the methods section entitled: Human cerebrospinal fluid cells.

REVIEWER COMMENTS

Reviewer #3 (Remarks to the Author):

I thank the authors for making these important clarifications to the statistical analyses. From my point of view, the manuscript is now ready for publication.

Reviewer #4 (Remarks to the Author):

I appreciate the authors for their efforts in addressing my comments. While numerous inquiries have been resolved, two principal concerns persist:

1. Validation experiments, encompassing the assessment of cytokine profiles at the protein level and image analysis techniques such as immunohistochemistry, are warranted to substantiate the discernible disparity between Nur77-GFP high and low cell populations. Notably, the authors have demonstrated proficiency in isolating these distinct cell types, rendering it imperative to undertake an examination of the differential expression of pivotal molecular entities. I strongly advocate for the inclusion of such investigations.

2. In order to corroborate the therapeutic efficacy of cell transfer, it is imperative for the authors to present histopathological data. The absence of such image data impedes the elucidation of the enduring impact of cell transfer, particularly in the context of marginal CFU differences. Hence, the provision of histopathological evidence is imperative for a comprehensive understanding of the long-term ramifications of cell transfer.

Response to reviewers and new manuscript revisions are shown in GREEN:

Reviewer #4 (Remarks to the Author):

I appreciate the authors for their efforts in addressing my comments. While numerous inquiries have been resolved, two principal concerns persist:

1. Validation experiments, encompassing the assessment of cytokine profiles at the protein level and image analysis techniques such as immunohistochemistry, are warranted to substantiate the discernible disparity between Nur77-GFP high and low cell populations. Notably, the authors have demonstrated proficiency in isolating these distinct cell types, rendering it imperative to undertake an examination of the differential expression of pivotal molecular entities. I strongly advocate for the inclusion of such investigations.

We thank the reviewer and appreciate their interest in assessing cytokine profiles of the Nur77-GFP HI and LO populations. However, as we discussed in our prior response, in this manuscript we have focused on the transcriptional heterogeneity of effector CD4 T cells activated in vivo by antigen stimulation in the lungs, characterizing their localization, antigen specificity, surface marker phenotype, and TCR repertoire.

Notably, we have shown protein level data on the expression of key surface markers that strongly validates our bulk and single cell RNA-Seq transcriptional data, specifically around the key conclusions of our manuscript, which regard the differential expression of the costimulatory receptor *Tnfrsf4/OX40*. Importantly, in contrast to cytokine production, assessment of cell surface markers does require ex vivo restimulation.

This is an essential technical point for the design of our study: we cannot compare cytokine production without ex vivo restimulation, which would activate both Nur77-GFPLO and HI cells, making both Nur77-GFPHI. This technique is contrary to our goal to identify and characterize the subset of cells that recently received antigen receptor stimulation in vivo, at the site of Mtb infection. We feel that our transcriptional data on cytokine gene expression gives important information about the functional cytokine capacities of these two different cell subsets, even though it is not possible to assess cytokine protein expression within the constraints of our approach.

We appreciate the reviewer calling attention to this technical issue, and we have revised this version of the manuscript to clearly state and call greater attention to this limitation of our study.

2. In order to corroborate the therapeutic efficacy of cell transfer, it is imperative for the authors to present histopathological data. The absence of such image data impedes the elucidation of the enduring impact of cell transfer, particularly in the context of marginal CFU differences. Hence, the provision of histopathological evidence is imperative for a comprehensive understanding of the long-term ramifications of cell transfer.

We appreciate the reviewer's interest in histopathological assessments to corroborate and enhance the interpretation of our transcriptional, flow cytometry, CFU, and survival analyses.

Respectfully, given the central conclusions of our study about OX40 as a target for immunotherapy, we feel that a more essential question is the impact of OX40 agonism on histopathology, to improve understanding of the beneficial effect of this treatment. Based in part

on the reviewer's comments about this in their prior review, we have now completed these studies and are pleased to include them in supplemental figure S4 of the revised manuscript. We have added new sections of the Results and Methods to relate these experiments, as well as to the Discussion to contextualize these data in relation to our other findings.

These studies reviewed by a blinded veterinary pathologist, show that compared to isotype control antibody or PD-1 blockade, OX40 agonism significantly impacts lung inflammation, by quantitatively increasing the average size of areas of granulomatous inflammation, and qualitatively shifting the structure of each lesion to become less discretely organized, and inducing areas of pulmonary edema not seen with the other two treatments. Areas containing acid fast bacilli were not quantitatively changed in size, but qualitatively had more scattered and less discrete AF+ aggregates. We feel that these new data add an important corroborating dimension to our flow cytometry studies showing increases in CD4 T cell numbers and changes in their lung parenchymal localization in response to OX40 agonism but not PD-1 blockade.

REVIEWERS' COMMENTS

Reviewer #4 (Remarks to the Author):

I appreciate the authors for their efforts in addressing my comments. All of my inquiries have been resolved.

Response to reviewers are shown in RED:

Reviewer #4 (Remarks to the Author):

I appreciate the authors for their efforts in addressing my comments. All of my inquiries have been resolved.

We thank the reviewer for their constructive comments.